# Learning Physical Dynamics with Subequivariant Graph Neural Networks

**Jiaqi Han**
Tsinghua University

**Wenbing Huang**[*]
Gaoling School of Artificial Intelligence,
Renmin University of China
Beijing Key Laboratory of Big Data
Management and Analysis Methods

**Hengbo Ma**
University of California, Berkeley

**Jiachen Li**
Stanford University

**Joshua B. Tenenbaum**
MIT BCS, CBMM, CSAIL

**Chuang Gan**
UMass Amherst
MIT-IBM Watson AI Lab

## Abstract

Graph Neural Networks (GNNs) have become a prevailing tool for learning physical dynamics. However, they still encounter several challenges: 1) Physical laws abide by symmetry, which is a vital inductive bias accounting for model generalization and should be incorporated into the model design. Existing simulators either consider insufficient symmetry, or enforce excessive equivariance in practice when symmetry is partially broken by gravity. 2) Objects in the physical world possess diverse shapes, sizes, and properties, which should be appropriately processed by the model. To tackle these difficulties, we propose a novel backbone, *Subequivariant Graph Neural Network*, which 1) relaxes equivariance to subequivariance by considering external fields like gravity, where the universal approximation ability holds theoretically; 2) introduces a new subequivariant object-aware message passing for learning physical interactions between multiple objects of various shapes in the particle-based representation; 3) operates in a hierarchical fashion, allowing for modeling long-range and complex interactions. Our model achieves on average over 3% enhancement in contact prediction accuracy across 8 scenarios on Physion and $2\times$ lower rollout MSE on RigidFall compared with state-of-the-art GNN simulators, while exhibiting strong generalization and data efficiency. Code and videos are available at our project page: https://hanjq17.github.io/SGNN/.

## 1  Introduction

Learning and predicting the complicated physical dynamics and interactions between objects are vital to many tasks including physical reasoning [8, 38, 7, 19], scene understanding [3], and model-based planning and control [20, 23, 24, 37, 1]. Several learning-based differentiable simulators [31, 30, 22, 27] have been proposed, and they obtain promising achievements in simulating various kinds of interacting objects including rigid and fluids. The success mainly relies on the prevalence of graph neural networks [14, 30, 15], which are desirable tools for physical simulation, by modeling particles as nodes, physical relations as edges, and their interactions as the message passing thereon.

---

[*]Corresponding author: Wenbing Huang.

36th Conference on Neural Information Processing Systems (NeurIPS 2022).

Notably, physical world exhibits certain symmetries. For example, the way dominoes fall from left to right will exactly be preserved when the system is rotated horizontally to another direction. Inherently, humans reason about the dynamics with the preservation of such symmetry, and this inductive bias has been endorsed by the physical laws that also abide by the symmetries in our 3D world. Yet, regardless of this inductive bias, current differentiable simulators like GNS [30] and DPI [22] would fail to learn the real dynamics that can generalize to all directions. In the example of dominoes, if the data in training are positioned from left to right, GNS can predict well during testing when the dominoes also fall in the same direction, but performs poorly if the scene is rotated horizontally (see demo in the Supplementary Video). This observation implies that GNS overfits training samples without learning the true dynamics that obey the symmetry, limiting its generalization.

With this consideration, there have been a number of works, named geometrically equivariant graph neural networks [16], that leverage symmetry as an inductive bias in learning to simulate. These models are designed such that their outputs will rotate/translate/reflect in the same way as the inputs, hence retaining the symmetry. However, models like EGNN [32] and GMN [17] are exerted E(3)-equivariance, the full symmetry of the 3D Euclidean space. Such constraint is so strong that it penalizes all directions in the 3D space, which cannot be applied to scenarios with external fields, like gravity. The existence of gravity breaks the symmetry in the vertical direction, reducing E(3) to its subgroup. We formally characterize this phenomenon of equivariance relaxation as *subequivariance*.

In general, simulating physics on datasets like Physion [4] are highly challenging, due to the scale of the system (on average thousands of particles per system), the diversity of the interactions (*e.g.*, collision, friction, gravity), as well as multiple shapes, materials, or even rigidness of the objects. These factors require unique efforts in designing the simulators, which brings another weakness of current GNN-based methods: From a practical point of view, they seldom explicitly involve the geometric object information into message passing. Moreover, when modeling multiple interacting objects of different shapes, the interactions between or within objects are usually different. For example, the former aims to exchange momentum or energy across objects, while the latter usually accounts for the geometrical constraint. It is thus important to involve objectness to distinguish interactions between particles within objects from those across objects.

In this work, we propose Subequivariant Graph Neural Networks (SGNN) that consist of several features to tackle the above challenges. **1.** We relax equivariance to subequivariance and design subequivariant functions to physical scenarios with the existence of gravity, and excitingly, we have proved that our designed form inherits the approximation universality. **2.** We formulate a novel subequivariant object-aware message passing framework for modeling dynamics and interactions between objects of varying shapes. **3.** We incorporate the subequivariant message passing into a hierarchical model to deal with long-range and complicated object interactions. We demonstrate the efficacy of our SGNN for learning physical dynamics on a large-scale challenging dataset Physion with 8 different scenarios, and a 3-object RigidFall dataset. Experimental results show that our model is capable of yielding more accurate dynamics prediction, is highly data-efficient, and has strong generalization compared with the state-of-the-art learning-based differentiable physical simulators.

## 2  Related Work

**GNN-based physical dynamics simulators.** There have been many works that employ graph neural networks as physical simulators of dynamical systems [2, 26, 18]. Graph Network Simulator (GNS) proposed by [30] has been showcased a simple yet powerful tool in simulating large systems in particle-based representation of rigid and fluids by dynamically constructing interaction graph and performing multiple steps of information propagation. DPI [22] adds one-level of hierarchy to the rigid and predicts the rigid transformation via generalized coordinates, with applications to manipulation. There are also works that incorporate strong physical priors like Hamiltonian mechanics [29, 39], geometrical constraints [28] and shapes [27]. Despite the promising empirical results, these works have not given sufficient considerations on symmetry especially when external force like gravity presents, leaving room for enhancing their generalization to unseen testing data.

**Geometrically equivariant graph neural networks.** Equivariant graph neural networks [16] are a family of GNNs that are specifically designed to meet the constraint of certain symmetry, mostly involving translations, rotations, and/or reflections in Euclidean space. This goal is approached by several measures, including solving group convolution with irreducible representation [34, 12] or

leveraging invariant scalarization [35] like taking the inner product [32, 17]. Our approach also belongs to the scalarization family together with EGNN [32] and GMN [17]. Specifically, EGNN models the interaction with the invariant distance as input, computed as an inner product of the relative positions, and GMN generalizes to a multi-channel version by taking into consideration a stack of multiple vectors. These equivariant GNNs operate on particle graphs or point clouds, while we additionally consider a combination of both particle- and object-level information for physical simulation. More importantly, they are assumed a full Euclidean symmetry with strong equivariance constraint, while we elaborate how to relax the constraint in the existence of external fields like gravity by leveraging subequivariance.

**Equivariance on subgroups.** E($n$)-Steerable CNNs [36, 6] develop convolutional kernels that meet equivariance on E(3)/E(2) and their subgroups by leveraging restricted representation. EMLP [10] obtains equivariance on arbitrary matrix groups. However, these approaches require specifying the particular group/subgroup to perform equivariance, while our goal here is to relax the equivariance on subgroups of E(3) by considering external force field, having more physical implications. Moreover, these works rely on computationally expensive operations like irreducible representation or solving group constraints, while our formulation resorts to scalarization, which is easy to implement, efficient to compute, and also comes with necessary universality guarantee.

# 3 Subequivariant Graph Neural Networks

## 3.1 Background

**GNN-based simulators.** We start by introducing the mechanism of GNN-based simulators for learning physical dynamics. We consider the particle-based representation of a physical system with $N$ particles consisting of $M$ objects. At time $t$, each particle within the system possesses some state information, including **1.** the position $\vec{x}_i^{(t)} \in \mathbb{R}^3$ and the velocity $\vec{v}_i^{(t)} \in \mathbb{R}^3$, which are both directional vectors; **2.** some attributes $\boldsymbol{h}_i \in \mathbb{R}^n$ without geometric context, such as the rigidness; **3.** the dynamic spatial connections with other particles, where an edge will be constructed if the distance between two particles is smaller than a threshold $r$, namely, $\mathcal{E}^{(t)} = \{(i,j) : \|\vec{x}_i^{(t)} - \vec{x}_j^{(t)}\|_2 < r\}$. We denote the object category of particle $i$ as $o(i) \in \mathbb{Z}$ satisfying $o(i) = k$ if particle $i$ belongs to object $k$. The goal here is to predict the position of the next step $\vec{x}_i^{(t+1)}$ given the above system information at time $t$ , which can be favorably modeled by GNNs, *i.e.*, $\vec{x}_i^{(t+1)} = \varphi_{\text{GNN}} \left( \{\vec{x}_i^{(t)}\}, \{\vec{v}_i^{(t)}\}, \{\boldsymbol{h}_i\}, \mathcal{E}^{(t)} \right)$. Since the prediction at different time shares the same model, we will henceforth omit the temporal superscript $t$ for all variables for brevity.

The conventional GNN simulators $\varphi_{\text{GNN}}$ [22, 30, 26] offer a favorable solution by leveraging message-passing on the interaction graph, which computes:

$$\boldsymbol{m}_{ij} = \phi\left(\vec{\boldsymbol{x}}_i, \vec{\boldsymbol{v}}_i, \vec{\boldsymbol{x}}_j, \vec{\boldsymbol{v}}_j, \boldsymbol{h}_i, \boldsymbol{h}_j\right), \tag{1}$$

$$\vec{\boldsymbol{x}}_i', \vec{\boldsymbol{v}}_i', \boldsymbol{h}_i' = \psi\left(\sum_{j \in \mathcal{N}(i)} \boldsymbol{m}_{ij}, \vec{\boldsymbol{x}}_i, \vec{\boldsymbol{v}}_i, \boldsymbol{h}_i\right), \tag{2}$$

where $\mathcal{N}(i) = \{j : (i,j) \in \mathcal{E}\}$ is the neighbors of node $i$, and $\phi, \psi$ are the edge message function and node update function, respectively. The prediction is obtained by conducting several iterations of message passing. Nevertheless, such form does not guarantee the desirable symmetry context with common choices of $\phi$ and $\psi$, and meanwhile the object information has not been elaborated, leaving room for a more exquisite message passing scheme for learning complex physical dynamics. Recent works such as EGNN [32] and GMN [17] have actually considered E(3)-equivariance in the design of the functions $\phi$ and $\psi$ to pursue symmetry. However, they are not applicable to the case when symmetry is partially violated by gravity, which motivates our proposal in § 3.2. Prior to introducing our work, we first provide necessary preliminaries related to equivariance.

**Equivariance.** In this paper, we basically focus on equivariance in terms of E(3) transformations: translation, rotation, and reflection. The formal definition is provided below.

**Definition 1** (Equivariance and Invariance). *We call that the function $f : \mathbb{R}^{3 \times m} \times \mathbb{R}^n \to \mathbb{R}^{3 \times m'}$ is E(3)-equivariant, if for any transformation $g$ in E(3), $f(g \cdot \vec{\boldsymbol{Z}}, \boldsymbol{h}) = g \cdot f(\vec{\boldsymbol{Z}}, \boldsymbol{h})$, $\forall \vec{\boldsymbol{Z}} \in \mathbb{R}^{3 \times m}$, $\forall \boldsymbol{h} \in \mathbb{R}^n$. Similarly, $f$ is invariant if $f(g \cdot \vec{\boldsymbol{Z}}, \boldsymbol{h}) = f(\vec{\boldsymbol{Z}}, \boldsymbol{h})$, $\forall g \in E(3)$.*

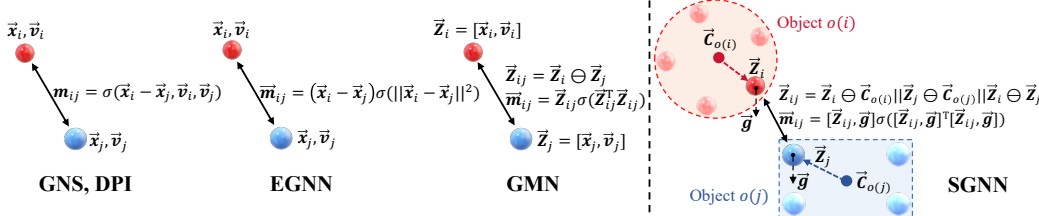

Figure 1: Comparison between different message passing. We omit the scalars $\boldsymbol{h}_i$ for simplicity.

In Definition 1, the group action $\cdot$ is instantiated as $g \cdot \vec{\boldsymbol{Z}} \coloneqq \boldsymbol{O}\vec{\boldsymbol{Z}}$ for the orthogonal transformation (rotation and reflection) where $\boldsymbol{O} \in O(3) \coloneqq \{\boldsymbol{O} \in \mathbb{R}^{3\times3} | \boldsymbol{O}^\top \boldsymbol{O} = \boldsymbol{I}\}$, and $g \cdot \vec{\boldsymbol{Z}} \coloneqq \vec{\boldsymbol{Z}} + \boldsymbol{t}$ for translation where $\boldsymbol{t} \in \mathbb{R}^3$. Note that for the input of $f$, we have added the the right-arrow superscript on $\vec{\boldsymbol{Z}}$ to distinguish it from the scalar $\boldsymbol{h}$ that is unaffected by the transformation, akin to our notations for the position $\vec{\boldsymbol{x}}_i$, velocity $\vec{\boldsymbol{v}}_i$, and attribute $\boldsymbol{h}_i$.

It is non-trivial to derive equivariant function particularly for orthogonal transformations. GMN [17] proposes a multichannel scalarization form, which yields,

$$f(\vec{\boldsymbol{Z}}, \boldsymbol{h}) \coloneqq \vec{\boldsymbol{Z}}\boldsymbol{V}, \quad \text{s.t.}\, \boldsymbol{V} = \sigma(\vec{\boldsymbol{Z}}^\top \vec{\boldsymbol{Z}}, \boldsymbol{h}), \tag{3}$$

where the inner product $\vec{\boldsymbol{Z}}^\top \vec{\boldsymbol{Z}} \in \mathbb{R}^{m\times m}$ is firstly computed and concatenated with $\boldsymbol{h}$, the resultant invariant term is then transformed by a function (usually a Multi-Layer Perceptron (MLP)) $\sigma : \mathbb{R}^{m\times m+n} \mapsto \mathbb{R}^{m\times m'}$ producing $\boldsymbol{V} \in \mathbb{R}^{m\times m'}$, and the directional output is acquired by taking a matrix multiplication of the basis $\vec{\boldsymbol{Z}}$ with $\boldsymbol{V}$. By joining the analyses in GMN along with [35], we have the universality of the formulation in Eq. (3), which is provided in Appendix A.

## 3.2 Subequivariant Object-aware Message Passing

To simultaneously benefit from symmetry and object-aware information, we develop Subequivariant Object-aware Message Passing (SOMP). It characterizes information aggregation between multi-channel vectors and scalars, while taking into account both the particle and object information. We use $\vec{\boldsymbol{Z}}_i \in \mathbb{R}^{3\times m}$ to indicate a stack of $m$ multi-channel 3D vectors; particularly, $\vec{\boldsymbol{Z}}_i$ is initialized as $[\vec{\boldsymbol{x}}_i, \vec{\boldsymbol{v}}_i], 1 \le i \le N$ by involving both the position and velocity. We additionally initialize the object features by pooling its particles: for object $k$, we have $\vec{\boldsymbol{C}}_k = \frac{1}{|\{i:o(i)=k\}|}\sum_{\{i:o(i)=k\}} \vec{\boldsymbol{Z}}_i, \boldsymbol{c}_k = \sum_{\{i:o(i)=k\}} \boldsymbol{h}_i, 1 \le k \le M$. We also introduce a binary operation "$\ominus$" which yields $\vec{\boldsymbol{Z}}_i \ominus \vec{\boldsymbol{Z}}_j = [\vec{\boldsymbol{Z}}_i - \vec{\boldsymbol{Z}}_j, \vec{\boldsymbol{v}}_i, \vec{\boldsymbol{v}}_j]$. It transforms the translation-equivariant vectors to be invariant and stacks those invariant vectors together, resulting in a translation-invariant representation while enriching the input vectors with more channels. The direction of gravity is set to be along the vertical axis.

In a high-level overview, we denote our message passing as the function $\varphi$ that updates each particle given the input states of all particles, objects, and graph connectivity:

$$\{(\vec{\boldsymbol{Z}}_i', \boldsymbol{h}_i')\}_{i=1}^N = \varphi\left(\{(\vec{\boldsymbol{Z}}_i, \boldsymbol{h}_i)\}_{i=1}^N, \{(\vec{\boldsymbol{C}}_k, \boldsymbol{c}_k)\}_{k=1}^M, \mathcal{E}\right). \tag{4}$$

Specifically, $\varphi$ is unfolded as the following message passing and aggregation computations:

$$\vec{\boldsymbol{Z}}_{ij} = (\vec{\boldsymbol{Z}}_i \ominus \vec{\boldsymbol{C}}_{o(i)}) \| (\vec{\boldsymbol{Z}}_j \ominus \vec{\boldsymbol{C}}_{o(j)}) \| (\vec{\boldsymbol{Z}}_i \ominus \vec{\boldsymbol{Z}}_j), \tag{5}$$

$$\boldsymbol{h}_{ij} = \boldsymbol{h}_i \| \boldsymbol{c}_{o(i)} \| \boldsymbol{h}_j \| \boldsymbol{c}_{o(j)}, \tag{6}$$

$$\vec{\boldsymbol{M}}_{ij}, \boldsymbol{m}_{ij} = \phi_{\vec{\boldsymbol{g}}}\left(\vec{\boldsymbol{Z}}_{ij}, \boldsymbol{h}_{ij}\right), \tag{7}$$

$$(\vec{\boldsymbol{Z}}_i', \boldsymbol{h}_i') = (\vec{\boldsymbol{Z}}_i, \boldsymbol{h}_i) + \psi_{\vec{\boldsymbol{g}}}\left((\sum_{j\in\mathcal{N}(i)} \vec{\boldsymbol{M}}_{ij}) \| (\vec{\boldsymbol{Z}}_i \ominus \vec{\boldsymbol{C}}_{o(i)}), (\sum_{j\in\mathcal{N}(i)} \boldsymbol{m}_{ij}) \| \boldsymbol{h}_i \| \boldsymbol{c}_{o(i)}\right), \tag{8}$$

where $\|$ is the concatenation along the channel dimension, $\mathcal{N}(i) = \{j : (i, j) \in \mathcal{E}\}$ is the neighbors of node $i$, and both $\phi_{\vec{\boldsymbol{g}}}$ and $\psi_{\vec{\boldsymbol{g}}}$ are subequivariant and will be defined in detail in Eq. (9). In detail, we first derive the multi-channel vector input $\vec{\boldsymbol{Z}}_{ij}$ for the interaction between particle $i$ and $j$ in Eq. (5)

by stacking the three terms along the channel dimension, including the relative information of the particle with respect to its belonged object: $\vec{\boldsymbol{Z}}_i \ominus \vec{\boldsymbol{C}}_{o(i)}$ and $\vec{\boldsymbol{Z}}_i \ominus \vec{\boldsymbol{C}}_{o(i)}$, and the relative information between particles $\vec{\boldsymbol{Z}}_i \ominus \vec{\boldsymbol{Z}}_j$. By this means, we arrive at a translation-invariant representation $\vec{\boldsymbol{Z}}_{ij}$ with rich object-aware geometric information. For the invariant features we simply concatenate them in Eq. (6). Subsequently, in Eq. (7) we feed the interactions $\vec{\boldsymbol{Z}}_{ij}$ and $\boldsymbol{h}_{ij}$ into the subequivariant message function $\phi_{\vec{\boldsymbol{g}}}$, yielding the vector message $\vec{\boldsymbol{M}}_{ij}$ and scalar message $\boldsymbol{m}_{ij}$[2]. Finally, Eq. (8) first performs message aggregation and then updates the states by another subequivariant function $\psi_{\vec{\boldsymbol{g}}}$, obtaining the eventual result. With the updated $\vec{\boldsymbol{Z}}'_i$ and $\boldsymbol{h}'_i$, one can readily obtain the output with equivariance or invariance as desired, which, in our case, implies $\vec{\boldsymbol{Z}}'_i = [\vec{\boldsymbol{x}}'_i]$ by setting $m' = 1$.

**Comparison with existing GNN-based simulators.** The core of satisfying equivariance lies in taking the invariant inner product before feeding the vectors into the MLP. Such design shares a similar spirit in nature with existing works including EGNN [32] and GMN [17]. The interaction modeling of EGNN only considers the relative position $\vec{\boldsymbol{x}}_i - \vec{\boldsymbol{x}}_j$ and its inner product $\|\vec{\boldsymbol{x}}_i - \vec{\boldsymbol{x}}_j\|^2$, which might have limited expressivity while tackling complex physical interactions and dynamics between objects. GMN further extends to a multi-channel interaction $\vec{\boldsymbol{Z}}_i \ominus \vec{\boldsymbol{Z}}_j$ where $\vec{\boldsymbol{Z}}_i = [\vec{\boldsymbol{x}}_i, \vec{\boldsymbol{v}}_i]$. Nevertheless, it lacks the necessary object information, while we explicitly involve the particle-object correlation $\vec{\boldsymbol{Z}}_i \ominus \vec{\boldsymbol{C}}_{o(i)}$ in our SOMP. Besides, GNS [30] and DPI [22] only enforces translation equivariance by directly feeding $[\vec{\boldsymbol{x}}_i - \vec{\boldsymbol{x}}_j, \vec{\boldsymbol{v}}_i, \vec{\boldsymbol{v}}_j]$ into an MLP without taking the inner product. They fail to preserve the O(3)-equivariance and consequently have weaker generalization as we will illustrate in our experiments. Fig. 1 summarizes and compares the message-passing schemes of GNS, EGNN, GMN and our SOMP. Particularly, our design of SOMP takes careful considerations: **1.** Equivariance is still permitted with both geometric and scalar object features involved; **2.** The object information can also be constantly updated during message passing; **3.** The expressivity of SOMP is enhanced over EGNN and GMN with object information considered. We provide detailed theoretical comparisons in Appendix A.2 by showing that EGNN and GMN are indeed special cases of SGNN.

We now present the detailed formulations of $\phi_{\vec{\boldsymbol{g}}}$ in Eq. (7) and $\psi_{\vec{\boldsymbol{g}}}$ in Eq. (8). Since the full symmetry is violated by gravity $\vec{\boldsymbol{g}} \in \mathbb{R}^3$, and the dynamics of the system will naturally preserve a gravitational acceleration in the vertical direction. By this means, the orthogonal symmetry is no longer maintained in every direction but only restricted to the subgroup $O_{\vec{\boldsymbol{g}}}(3) := \{\boldsymbol{O} \in O(3) \mid \boldsymbol{O}\vec{\boldsymbol{g}} = \vec{\boldsymbol{g}}\}$, that is, the rotations/reflections around the gravitational axis. We term such a reduction of equivariance as a novel notion: *subequivariance*. To reflect this special symmetry, we augment Eq. (3) by:

$$f_{\vec{\boldsymbol{g}}}(\vec{\boldsymbol{Z}}, \boldsymbol{h}) = [\vec{\boldsymbol{Z}}, \vec{\boldsymbol{g}}]\boldsymbol{V}_{\vec{\boldsymbol{g}}}, \quad \text{s.t.} \boldsymbol{V}_{\vec{\boldsymbol{g}}} = \sigma([\vec{\boldsymbol{Z}}, \vec{\boldsymbol{g}}]^\top[\vec{\boldsymbol{Z}}, \vec{\boldsymbol{g}}], \boldsymbol{h}), \tag{9}$$

where $\sigma : \mathbb{R}^{(m+1)\times(m+1)} \to \mathbb{R}^{(m+1)\times m'}$ is an MLP. Compared with Eq. (3), here we just augment the directional input with $\vec{\boldsymbol{g}}$. Interestingly, such a simple augmentation is universally expressive:

**Theorem 1.** *Let $f_{\vec{\boldsymbol{g}}}(\vec{\boldsymbol{Z}}, \boldsymbol{h})$ be defined by Eq. (9). Then, $f_{\vec{\boldsymbol{g}}}$ is $O_{\vec{\boldsymbol{g}}}(3)$-equivariant. More importantly, For any $O_{\vec{\boldsymbol{g}}}(3)$-equivariant function $\hat{f}(\vec{\boldsymbol{Z}}, \boldsymbol{h})$, there always exists an MLP $\sigma$ satisfying $\|\hat{f} - f_{\vec{\boldsymbol{g}}}\| < \epsilon$ for arbitrarily small positive value $\epsilon$.*

The proof is non-straightforward and deferred to Appendix A.1. The detailed architectural view of the proposed SOMP is depicted in Fig. 8 in Appendix A.3. We leverage our designed subequivariant function with such desirable properties for $\phi_{\vec{\boldsymbol{g}}}$ and $\psi_{\vec{\boldsymbol{g}}}$ in our subequivariant object-aware message passing. We immediately have the following theorem guaranteeing the validity of our design.

**Theorem 2.** *The message passing $\varphi$ (Eq. 4) is $O_{\vec{\boldsymbol{g}}}(3)$-equivariant.*

### 3.3 Application to Physical Scenes with Hierarchical Modeling

This subsection introduce the entire architecture of our subequivariant GNN. Many physical scenes are complicated, possibly involving contact, collision, and friction amongst multiple objects. In light of this, we further incorporate our message passing into a multi-stage hierarchical modeling framework. One of our interesting findings here is the edge separation. This is indeed motivated by the consideration that the interactions *between* or *within* objects are usually different, similar to

---

[2]For both $\phi_{\vec{\boldsymbol{g}}}$ and $\psi_{\vec{\boldsymbol{g}}}$, we expand more output channels besides $\boldsymbol{V}_{\vec{\boldsymbol{g}}}$ of $\sigma$ in Eq. (9), and assign it to the invariant message $\boldsymbol{m}_{ij}$ in Eq. (7) and $\boldsymbol{h}'_i$ in Eq. (8), respectively.

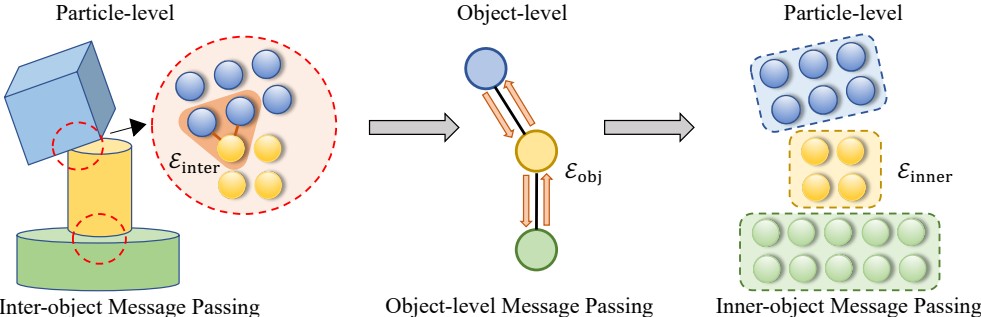

Figure 2: The overall architecture with hierarchical modeling, specified as three stages.

automorphism graph networks [9, 25, 33] where the neighboring objects formulate the isomorphism group of edges. The former serves as a bridge of exchanging momentum and energy via interaction forces, while the latter usually accounts for maintaining the rigid or plastic constraints. Therefore, it would be beneficial to disentangle these two kinds of interactions in message passing, which also distinguishes our hierarchical modeling from DPI [22]. The overall flowchart is provided in Fig. 2. To be specific, we alternate between the following three stages that implement $\varphi$ in Eq. (4) distinctly.

**1. Particle-level inter-object message passing.** We start by modeling the local interactions from the particle-level, where we only involve those between different objects. That is,

$$\{(\vec{\boldsymbol{Z}}_i', \boldsymbol{h}_i')\}_{i=1}^N = \varphi_1\left(\{(\vec{\boldsymbol{Z}}_i, \boldsymbol{h}_i)\}_{i=1}^N, \{(\vec{\boldsymbol{C}}_k, \boldsymbol{c}_k)\}_{k=1}^M, \mathcal{E}_{\text{inter}}\right), \tag{10}$$

where $\mathcal{E}_{\text{inter}} = \{(i,j) : (i,j) \in \mathcal{E}, o(i) \neq o(j)\}$.

**2. Object-level message passing.** Given the renewed particle-level information produced by the first stage, we are now ready to construct the object-level message passing graph. The object-level message passing is proceeded as

$$\{(\vec{\boldsymbol{C}}_k', \boldsymbol{c}_k')\}_{k=1}^M = \varphi_2\left(\{(\vec{\boldsymbol{C}}_k, \boldsymbol{c}_k)\}_{k=1}^M, \{\emptyset\}, \mathcal{E}_{\text{obj}}\right), \tag{11}$$

where $\mathcal{E}_{\text{obj}}$ comprises the edges between interacted objects, indicating $\mathcal{E}_{\text{obj}} = \{(k,l) : \exists (i,j) \in \mathcal{E}_{\text{inter}}, o(i) = k, o(j) = l\}$. As a specific instantiation of $\varphi$ in Eq. (4), here each object is considered as a node and the original object information in Eq. (5-8) is omitted (this is why we denote the second input of $\varphi_2$ as the empty set $\emptyset$). Furthermore, we leverage a pooling of the updated particle-level information from the first stage as the object-level interactions, or formally, we re-define the calculation of "$\ominus$" for two different objects indexed by $k, l$ as $\vec{\boldsymbol{C}}_k \ominus \vec{\boldsymbol{C}}_l = \text{Mean-Pool}_{(i,j)\in\mathcal{E}_{\text{inter}}}\left(\vec{\boldsymbol{Z}}_i' \ominus \vec{\boldsymbol{Z}}_j'\right)$, and similarly, $\boldsymbol{c}_k \| \boldsymbol{c}_l = \text{Mean-Pool}_{(i,j)\in\mathcal{E}_{\text{inter}}}\left(\boldsymbol{h}_i' \| \boldsymbol{h}_j'\right)$.

**3. Particle-level inner-object message passing.** Finally, given the updated object state information, we carry out the message passing for the particles only along the inner-object edges $\mathcal{E}_{\text{inner}} = \{(i,j) : (i,j) \in \mathcal{E}, o(i) = o(j)\}$, namely,

$$\{(\vec{\boldsymbol{Z}}_i'', \boldsymbol{h}_i'')\}_{i=1}^N = \varphi_3\left(\{(\vec{\boldsymbol{Z}}_i, \boldsymbol{h}_i)\}_{i=1}^N, \{(\vec{\boldsymbol{C}}_k', \boldsymbol{c}_k')\}_{k=1}^M, \mathcal{E}_{\text{inner}}\right). \tag{12}$$

We use $\{\vec{\boldsymbol{Z}}_i''\}_{i=1}^N$ as the proposal of the positions in the next time step, akin to the way in § 3.2.

## 4 Experiments

We conduct evaluations on Physion [4] and RigidFall [21]. Physion [4] is a large scale dataset created by the ThreeDWorld simulator [13], which consists of eight different scenarios, including Dominoes, Contain, Collide, Drop, Roll, Link, Support, and Drape. These scenarios possess diverse physical scenes with complicated object interactions, serving as a challenging benchmark for evaluating dynamics models. Particularly, Drape also involves simulating *deformable* objects. RigidFall [21] is a simulation dataset whose scenes involve several cubes falling and colliding under varying magnitude of gravitational acceleration.

Table 1: Contact prediction accuracy (%) on Physion. Results are averaged across 3 runs.

| | Dominoes | Contain | Link | Drape | Support | Drop | Collide | Roll |
|---|---|---|---|---|---|---|---|---|
| GNS* [30] | $78.6_{\pm0.9}$ | $71.6_{\pm1.6}$ | $66.7_{\pm1.5}$ | $58.8_{\pm1.0}$ | $68.2_{\pm1.6}$ | $65.3_{\pm1.1}$ | $\mathbf{86.1}_{\pm0.5}$ | $81.3_{\pm1.8}$ |
| DPI* [22] | $82.3_{\pm1.3}$ | $72.3_{\pm1.8}$ | $63.7_{\pm2.2}$ | $53.3_{\pm0.9}$ | $64.8_{\pm2.0}$ | $70.7_{\pm0.8}$ | $84.4_{\pm0.7}$ | $82.3_{\pm0.6}$ |
| GNS [30] | $53.3_{\pm1.8}$ | $70.7_{\pm2.2}$ | $58.0_{\pm1.7}$ | $58.0_{\pm1.3}$ | $61.3_{\pm1.7}$ | $65.0_{\pm0.7}$ | $76.7_{\pm1.2}$ | $80.0_{\pm0.4}$ |
| DPI [22] | $57.3_{\pm2.2}$ | $71.9_{\pm1.3}$ | $63.0_{\pm1.9}$ | $52.3_{\pm1.2}$ | $58.0_{\pm1.1}$ | $68.7_{\pm0.7}$ | $78.7_{\pm1.3}$ | $80.5_{\pm0.8}$ |
| GNS-Rot [30] | $74.7_{\pm1.2}$ | $72.7_{\pm2.0}$ | $63.1_{\pm2.1}$ | $57.3_{\pm1.0}$ | $65.5_{\pm1.3}$ | $64.7_{\pm1.3}$ | $84.0_{\pm1.0}$ | $79.4_{\pm2.1}$ |
| DPI-Rot [22] | $72.7_{\pm1.7}$ | $69.4_{\pm2.1}$ | $65.2_{\pm2.3}$ | $52.8_{\pm1.0}$ | $66.7_{\pm0.8}$ | $72.3_{\pm0.5}$ | $83.2_{\pm1.6}$ | $80.1_{\pm0.5}$ |
| EGNN [32] | $61.3_{\pm1.1}$ | $66.0_{\pm1.9}$ | $52.7_{\pm1.2}$ | $54.7_{\pm0.8}$ | $60.0_{\pm0.5}$ | $63.3_{\pm0.9}$ | $76.7_{\pm1.4}$ | $79.8_{\pm0.4}$ |
| EGNN-S | $72.0_{\pm2.1}$ | $64.6_{\pm1.8}$ | $55.3_{\pm2.1}$ | $55.3_{\pm1.0}$ | $60.5_{\pm1.5}$ | $69.3_{\pm2.0}$ | $79.3_{\pm1.5}$ | $81.6_{\pm0.9}$ |
| GMN [17] | $54.7_{\pm0.5}$ | $57.6_{\pm2.2}$ | $54.5_{\pm1.6}$ | $57.6_{\pm1.3}$ | $55.1_{\pm1.6}$ | $54.2_{\pm1.1}$ | $79.5_{\pm1.5}$ | $81.3_{\pm1.3}$ |
| GMN-S | $55.6_{\pm1.3}$ | $65.3_{\pm1.7}$ | $55.1_{\pm2.8}$ | $57.0_{\pm1.8}$ | $59.3_{\pm2.2}$ | $57.3_{\pm1.5}$ | $81.2_{\pm1.2}$ | $79.3_{\pm1.5}$ |
| Our SGNN | $\mathbf{89.1}_{\pm1.5}$ | $\mathbf{78.1}_{\pm1.5}$ | $\mathbf{73.3}_{\pm1.1}$ | $\mathbf{60.6}_{\pm0.5}$ | $\mathbf{71.2}_{\pm0.9}$ | $\mathbf{74.3}_{\pm1.0}$ | $85.3_{\pm1.1}$ | $\mathbf{84.2}_{\pm0.6}$ |

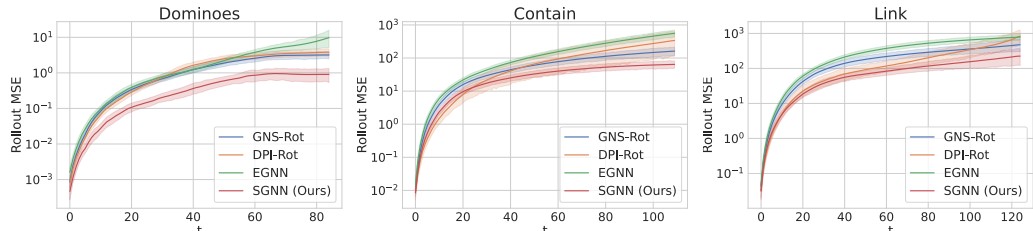

Figure 3: The rollout-MSE curves on Dominoes, Link, and Contain. Our model achieves the lowest MSE. Colored areas indicate 95% confidence intervals. Full results in Appendix C.2.

## 4.1 Baselines

We compare SGNN with several SOTA particle-based GNN simulators, including two non-equivariant models GNS [30] and DPI [22], as well as two equivariant models EGNN [32] and GMN [17]. Notably, the data preprocessing provided by Physion [4] for particle-based methods does not consider the camera-angle for the rotation of 3D coordinates, leading to the situation that in some scenarios, *e.g.*, Dominoes, Link, Support, and Collide, there exist a directional bias (from left to right horizontally) in the trajectories. This motivates us to consider three evaluation protocols: **1.** Use the original training and testing set with directional bias. **2.** Train on the original set and test on the randomly-rotated testing set. **3.** Train on the randomly-rotated training set (akin to applying data augmentation of rotations [5]), and test on the randomly-rotated testing set. Here all rotations are restricted to the ones around gravity. We dub the results obtained from the three cases GNS*, GNS, and GNS-Rot for GNS, and similarly for DPI. For EGNN and GMN, we also propose to adapt them to be subequivariant (EGNN-S and GMN-S) by adding an extra vector along the gravity in their update of velocities. Details are in Appendix B. Note that EGNNs, GMNs, and our SGNN always produce exactly the same result in the three scenarios due to their equivariance (or subequivariance). For RigidFall, such bias is not observed since the cubes are falling vertically driven by gravity.

## 4.2 Evaluation on Physion

**Experimental setup.** We strictly follow the training and evaluation protocol proposed in [4] for the particle-based simulators. We employ two evaluation metrics: **1.** Contact prediction accuracy. This metric, as also adopted by [4], evaluates whether two targeted objects labeled in the dataset will contact during the model rollout. **2.** Rollout MSE, the mean squared error between model rollout and the ground truth. We reuse the training protocol and hyper-parameters suggested in Physion for baselines and our model to ensure a fair comparison. In detail, all MLPs are initialized with 3 projection layers and a hidden dimension of 200. The networks are trained with an Adam optimizer, using an initial learning rate 0.0001 and an early-stopping of 10 epochs on the validation loss. We use 4 iterations in each message passing of our model. We run RANSAC [11] during testing on all models to help enforce the rigid constraint as suggested by Physion. More experimental details including a comparison of computational complexity are deferred to Appendix B.

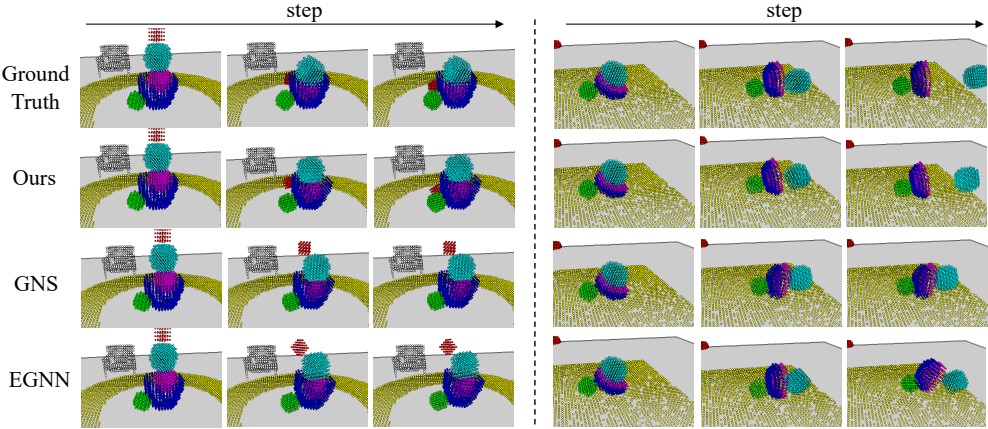

Figure 4: Visualization examples on Physion. Our model yields accurate long-term predictions.

**Results.** We present the average contact prediction accuracy with std in Table 1, and the rollout MSE in Fig. 3. The detailed MSE on all 8 scenarios is deferred to Appendix C.2 due to space limit. We interpret our results by answering the following questions. **Q1.** *Is maintaining the symmetry important to models that learn to simulate?* For the non-equivariant models GNS and DPI, their performance usually encounters a significant drop if they are trained on the original data but tested on the rotated, particularly on the scenarios with bias in directions. For example, the accuracy of GNS drops from 78.6% to 53.3% on Dominoes when comparing GNS* with GNS. Such observation also holds for DPI and on more scenarios like Collide and Support. Using data augmentation would help in relieving the issue, but is still hard to recover the performance. The accuracy on Dominoes yielded by GNS-Rot is 74.7%, still worse than 78.6% by GNS*. Our SGNN instead leverages subequivariance to generalize to all horizontal directions, and is always guaranteed to produce the same prediction regardless of any rotations around gravity, and there is also no need to apply any data augmentation of rotations during training. Interestingly, EGNN-S and GMN-S offers improvements over EGNN and GMN, implying the necessity of subequivariance, but are still outperformed by SGNN by a large margin. **Q2.** *Does SGNN learn the physics and simulate the dynamics better than other models?* As displayed in Table 1, SGNN achieves the highest prediction accuracy on 7 out of the 8 scenarios, and is also very competitive on Collide. The improvements are very significant (>5% accuracy enhancement over the second best) on scenarios including Dominoes, Contain, and Link, where there are multiple objects contacting, colliding, and interacting. This observation is also supported by the rollout MSE curves in Fig. 3, showing our SGNN consistently yielding lower prediction error along the trajectories. **Q3.** *How does SGNN perform compared with other equivariant models?* For EGNN and GMN, enforcing such a strong constraint of E(3)-equivariance leads to inferior performance in scenes with gravity. For instance, in Dominoes, Contain, and Link, the accuracy of EGNN and GMN only stays around 60%, while SGNN achieves promising results with subequivariant object-aware message passing.

**Generalization across scenarios.** We further evaluate the generalization capability of our model across various scenarios. To fulfill this goal, we leverage models trained in certain scenarios to test them in others. We summarize the results in Fig. 5, where the models are trained in the scenarios indexed by the rows and evaluated on those indexed by the columns. It is observed that our model exhibits significantly stronger generalization than GNS, the best-performed baseline on Physion. For example, our model, although trained with the dynamics of Contain or Link, still yields a high accuracy (>78%) when tested on Drop, showcasing that our model is more advantageous in learning the dynamics and interactions of various objects.

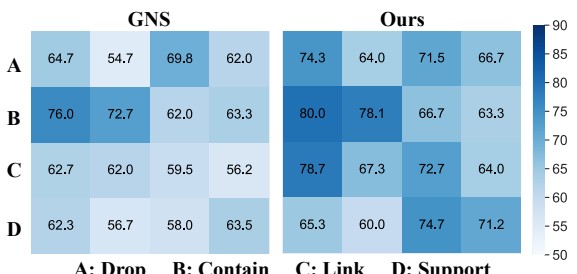

Figure 5: Generalization across tasks. Row/column indicates the training/testing scenario, respectively.

**Ablation studies.** Here we conduct several ablations to inspect how our proposed components contribute to the overall performance, and the results are given in Table 2. **1.** Subequivariance. We replace our subequivariant formulation of $\phi_{\vec{g}}$ and $\psi_{\vec{g}}$ by the E(3)-equivariant counterpart without gravity. It is clear that, in this way, the model is restricted by strong equivariance constraints, which leads to a significant degradation in performance. This verifies the necessity of

Table 2: Ablation studies.

|  | Dominoes | Contain | Drop |
|---|---|---|---|
| SGNN | 89.1 | 78.1 | 74.1 |
| E(3)-equivariance | 79.6 | 65.3 | 68.1 |
| w/o object-aware | 81.2 | 68.7 | 70.1 |
| w/o hierarchy | 83.5 | 72.5 | 71.3 |
| w/o edge separation | 85.0 | 74.8 | 72.0 |
| Steer-SE(2)-GNN [36] | 59.1 | 66.7 | 66.7 |

relaxing the full-equivariant model with our introduced method for physical scenarios with gravity. Moreover, we extend the Steerable E(2)-CNNs [36, 6] to GNNs as an alternative to obtain equivariance on the SE(2) subgroup (more details are results are in Appendix C.5). Although better than EGNN, the adapted Steerable SE(2)-GNN is still inferior to SGNN, implying our physics-inspired SOMP is more advantageous on simulating physical dynamics. **2.** Object-aware message passing. We replace $\vec{C}_k$ and $c_k$ in Eq. (10) by zeros, eliminating the object information from message passing. Without this necessary information, the model might fail to capture useful geometric vectors like $\vec{x}_i - \vec{x}_{o(i)}$, which is closely related to physical quantities like torque. **3.** Hierarchy. We compare our model with its counterpart without hierarchy. The flat model encounters an average of 4% drop in the prediction accuracy, showcasing the advantages of leveraging the particle- and object-level message passing for modeling complex object interactions in physical scenes. **4.** Edge separation. We employ different sets of edges in the inter-object and inner-object message passing. We argue that it relieves the learning complexity of the message-passing with different types of relations. It is observed that removing edge separation leads to detriment in the accuracy in various scenarios.

**Generalization toward other rotations.** To further investigate whether incorporating our subequivariance truly helps the model to learn the effect of gravity, we apply a rotation around a *non-gravity* axis, resulting in scenarios in Fig. 6 where dominoes are placed on an incline while gravity still points downwards vertically. Interestingly, SGNN well generalizes to this novel scenario and reasonably simulates the effect of gravity. Particularly, the domino at the bottom starts to slide down along the table driven by gravity. More visualizations are provided in the video.

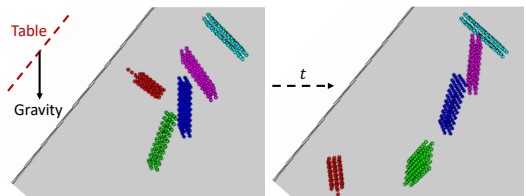

Figure 6: Rotation around a non-gravity axis.

**Qualitative comparisons.** We provide visualization samples on Physion in Fig. 4. More cases are shown by our Supplementary Video. Our model consistently yields accurate predictions across various scenarios.

## 4.3 Evaluation on RigidFall

Table 3: Rollout MSE ($\times 10^{-2}$) on RigidFall. |Train| denotes the number of training samples.

|  | |Train| = 200 | | |Train| = 500 | | |Train| = 1000 | | |Train| = 5000 | |
|---|---|---|---|---|---|---|---|---|
|  | $t = 20$ | $t = 40$ | $t = 20$ | $t = 40$ | $t = 20$ | $t = 40$ | $t = 20$ | $t = 40$ |
| GNS [30] | $2.40_{\pm 1.32}$ | $6.79_{\pm 3.43}$ | $1.89_{\pm 0.87}$ | $5.41_{\pm 2.78}$ | $1.09_{\pm 0.70}$ | $3.38_{\pm 2.15}$ | $0.65_{\pm 0.43}$ | $2.38_{\pm 1.66}$ |
| DPI [22] | $1.71_{\pm 0.87}$ | $5.37_{\pm 2.86}$ | $1.48_{\pm 0.78}$ | $4.67_{\pm 2.55}$ | $1.24_{\pm 0.72}$ | $3.97_{\pm 2.35}$ | $0.52_{\pm 0.36}$ | $2.40_{\pm 1.53}$ |
| EGNN [32] | $1.89_{\pm 1.34}$ | $5.66_{\pm 2.81}$ | $1.07_{\pm 0.71}$ | $3.94_{\pm 2.20}$ | $0.95_{\pm 0.44}$ | $2.72_{\pm 1.75}$ | $0.78_{\pm 0.45}$ | $2.63_{\pm 1.72}$ |
| GMN [17] | $2.74_{\pm 1.24}$ | $7.08_{\pm 3.37}$ | $2.50_{\pm 0.98}$ | $6.34_{\pm 3.11}$ | $1.91_{\pm 1.10}$ | $4.50_{\pm 2.21}$ | $1.36_{\pm 0.63}$ | $3.50_{\pm 1.98}$ |
| Our SGNN | $\mathbf{0.72}_{\pm 0.60}$ | $\mathbf{2.44}_{\pm 1.74}$ | $\mathbf{0.39}_{\pm 0.23}$ | $\mathbf{1.47}_{\pm 1.26}$ | $\mathbf{0.31}_{\pm 0.17}$ | $\mathbf{1.09}_{\pm 0.93}$ | $\mathbf{0.20}_{\pm 0.14}$ | $\mathbf{1.01}_{\pm 0.97}$ |

**Experimental setup.** We use the code provided by [22]. We record the rollout MSE at $t = 20$ and 40. To compare the data efficiency of different models, we consider four cases with the size of training set ranging from 200 to 5000. More details are in Appendix B.

**Results.** The results are presented in Table 3 and visualized in Fig. 7. Our model consistently yields the best predictions regardless of the time step $t$ and the size of training set. Notably, the rollout

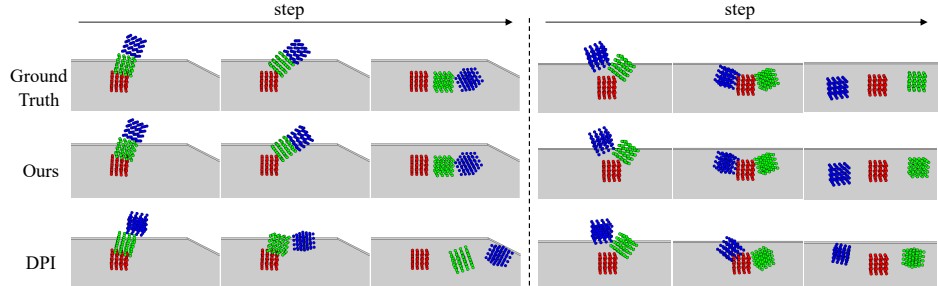

Figure 7: Visualization examples on RigidFall. Our SGNN provides accurate simulation trajectories.

error still maintains very low even if only 200 training samples are provided, verifying the strong data-efficiency and generalization of SGNN.

## 5 Discussion

**Limitations and Future Work.** Our model relies on the particle-based representation of the physical systems, which, in real-world scenarios, might be difficult to obtain, or usually with noise. Our model does not explicitly consider self-contact. Future directions include augmenting our dynamics model with strong physical priors like Hamiltonian [29, 39], or combining it with groundings of visual prior [21] with application to more sophisticated tasks like physical reasoning or robot manipulation.

**Conclusion.** We propose Subequivariant Graph Neural Networks for modeling physical dynamics of multiple interacting objects. We inject appropriate symmetry into a hierarchical message passing framework, and takes into account both particle- and object-level state messages. In particular, subequivariance is a novel concept for characterizing the relaxation of equivariance from a group to its subgroup. We show that relaxation from full equivariance to subequivariance can be applied to systems with gravity involved. Experiments verify that SGNN accurately captures the dynamics with the guarantee of the desirable symmetry, and exhibits strong generalization with high data-efficiency.

## Acknowledgments and Disclosure of Funding

We thank the anonymous reviewers for their constructive suggestions. This project was in part supported by MIT-IBM Watson AI Lab, Amazon Research Award Mitsubishi Electric. Dr. Wenbing Huang was supported by the following projects: the National Natural Science Foundation of China (No.62006137); Guoqiang Research Institute General Project, Tsinghua University (No. 2021GQG1012); Beijing Outstanding Young Scientist Program (No. BJJWZYJH012019100020098).

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
