# Supplementary Materials for
# Learning Physical Dynamics with Subequivariant Graph Neural Networks

## Contents

## A  Theoretical Preliminaries and Proofs

In the main paper, we have sketched the definitions and conclusions related to E(3) equivariance and subequivariance. Here, we introduce more details to facilitate the understanding for these conceptions.

We first recap the orthogonal group $O(3) = \{O \in \mathbb{R}^{3 \times 3} \mid O^\top O = I_3\}$ and the translation group $T(3) = \{t \in \mathbb{R}^3\}$. Then the Euclidean group is given by $E(3) = O(3) \ltimes T(3)$, where $\ltimes$ denotes the semidirect product. Basically, E(3) is a group of orthogonal transformations (rotations and reflections) and translations.

Definition 1 is rewritten as:

36th Conference on Neural Information Processing Systems (NeurIPS 2022).

**Definition 1** (Equivariance and Invariance). *We call that the function $f : \mathbb{R}^{3 \times m} \times \mathbb{R}^n \to \mathbb{R}^{3 \times m'}$ is G-equivariant, if for any transformation $g$ in a group $G$, $f(g \cdot \vec{Z}, h) = g \cdot f(\vec{Z}, h)$, $\forall \vec{Z} \in \mathbb{R}^{3 \times m}$, $\forall h \in \mathbb{R}^n$. Similarly, $f$ is invariant if $f(g \cdot \vec{Z}, h) = f(\vec{Z}, h)$, $\forall g \in G$.*

Eq. (3) in the main paper is repeated as:

$$f(\vec{Z}, h) := \vec{Z} V, \quad \text{s.t.} V = \sigma(\vec{Z}^\top \vec{Z}, h), \tag{13}$$

where the Multi-Layer Perceptron (MLP) function $\sigma : \mathbb{R}^{m \times m + n} \mapsto \mathbb{R}^{m \times m'}$ produces $V \in \mathbb{R}^{m \times m'}$. It is easy to justify that the function defined in Eq. (13) is O(3)-equivariant. Moreover, by joining the analyses in GMN along with the conclusion by [11], we immediately have the universality of the formulation in Eq. (13):

**Proposition 1** ([5, 11]). *Let $f$ be defined by Eq. (13). For any O(3)-equivariant function $\hat{f}(\vec{Z}, h)$, there always exists an MLP $\sigma$ satisfying $\|\hat{f} - f\| < \epsilon$ for arbitrarily small positive value $\epsilon$.*

To prove this theorem, we require the following two lemmas:

**Lemma 1.** *For any O(3)-equivariant function $\hat{f}(\vec{Z}, h)$, it must fall into the subspace spanned by the columns of $\vec{Z}$, namely, there exists a function $s(\vec{Z}, h)$, satisfying $\hat{f}(\vec{Z}, h) = \vec{Z} s(\vec{Z}, h)$.*

*Proof.* The proof is given by [11]. Essentially, suppose $\vec{Z}^\perp$ is the orthogonal complement of the column space of $\vec{Z}$. Then there must exit functions $s(\vec{Z}, h)$ and $s^\perp(\vec{Z}, h)$, satisfying $\hat{f}(\vec{Z}, h) = \vec{Z} s(\vec{Z}, h) + \vec{Z}^\perp s^\perp(\vec{Z}, h)$. We can always find an orthogonal transformation $O$ allowing $O\vec{Z} = \vec{Z}$ while $O\vec{Z}^\perp = -\vec{Z}^\perp$. With this transformation $O$, we have $\hat{f}(O\vec{Z}, h) = \hat{f}(\vec{Z}, h) = \vec{Z} s(\vec{Z}, h) + \vec{Z}^\perp s^\perp(\vec{Z}, h)$, and $O\hat{f}(\vec{Z}, h) = \vec{Z} s(\vec{Z}, h) - \vec{Z}^\perp s^\perp(\vec{Z}, h)$. The equivariance property of $\hat{f}$ implies $\vec{Z} s(\vec{Z}, h) + \vec{Z}^\perp s^\perp(\vec{Z}, h) = \vec{Z} s(\vec{Z}, h) - \vec{Z}^\perp s^\perp(\vec{Z}, h)$, which derives $s^\perp(\vec{Z}, h) = 0$. Hence, the proof is concluded. □

**Lemma 2.** *If the O(3)-equivariant function $\hat{f}(\vec{Z}, h)$ lies in the subspace spanned by the columns of $\vec{Z}$, then there exists a function $\sigma$ satisfying $\hat{f}(\vec{Z}, h) = \vec{Z}\sigma(\vec{Z}^\top Z, h)$.*

*Proof.* The proof is provided by Corollary 2 in [5]. The basic idea is that $\hat{f}(\vec{Z}, h)$ can be transformed to $\hat{f}(\vec{Z}, h) = \vec{Z}\eta(\vec{Z}, h)$ where $\eta(\vec{Z}, h)$ is O(3)-invariant. According to Lemma 1-2 in [5], $\eta(\vec{Z}, h)$ must be written as $\eta(\vec{Z}, h) = \sigma(\vec{Z}^\top \vec{Z}, h)$, which completes the proof. □

With Lemma 1-2, we always have $\hat{f}(\vec{Z}, h) = \vec{Z}\sigma(\vec{Z}^\top \vec{Z}, h)$. Since $\sigma$ can be universally approximated by MLP, then the conclusion in Proposition 1 is proved.

Nevertheless, the full symmetry is not always guaranteed and could be broken in certain directions owing to external force fields. For example, the existence of gravity breaks the symmetry by exerting a force field along the gravitational axis $\vec{g} \in \mathbb{R}^3$, and the dynamics of the system will naturally preserve a gravitational acceleration in the vertical direction. By this means, the orthogonal symmetry is no longer maintained in every direction but only restricted to the subgroup $O_{\vec{g}}(3) := \{O \in O(3) | O\vec{g} = \vec{g}\}$, that is, the rotations/reflections around the gravitational axis. We term such a reduction of equivariance as a novel notion: *subequivariance*.

**Definition 2** (Subequivariance and Subinvariance). *We call the function $f : \mathbb{R}^{3 \times m} \times \mathbb{R}^n \to \mathbb{R}^{3 \times m'}$ is O(3)-subequivariant induced by $\vec{g}$, if $f(O\vec{Z}, h) = Of(\vec{Z}, h)$, $\forall O \in O_{\vec{g}}(3)$, $\forall \vec{Z} \in \mathbb{R}^{3 \times m}$, $\forall h \in \mathbb{R}^n$; and similarly, it is O(3)-subinvariant induced by $\vec{g}$, if $f(O\vec{Z}, h) = f(\vec{Z}, h)$, $\forall O \in O_{\vec{g}}(3)$.*

Eq. (13) is clearly O(3)-subequivariant, but the O(3)-subequivariant function is unnecessarily the form like Eq. (13). Considering the example for the gravity itself which maps all particles to the direction $\vec{g}$. It is natural to see that the function $f$ by Eq. (13) fails to represent $\vec{g}$ if $\vec{g}$ does not fall into the subsubspace spanned by the columns of $\vec{Z}$. While this example provides a failure, it also inspires us to derive the following augmented version upon Eq. (13):

$$f_{\vec{g}}(\vec{Z}, h) = [\vec{Z}, \vec{g}] V_{\vec{g}}, \quad \text{s.t.} V_{\vec{g}} = \sigma([\vec{Z}, \vec{g}]^\top [\vec{Z}, \vec{g}], h), \tag{14}$$

where $\sigma : \mathbb{R}^{(m+1)\times(m+1)} \to \mathbb{R}^{(m+1)\times m'}$ is an MLP. Compared with Eq. (13), here we just augment the directional input with $\vec{g}$. Interestingly, such a simple augmentation is universally expressive, which is proved in the following section.

## A.1 Proof of Theorem 1

**Theorem 1.** *Let $f_{\vec{g}}(\vec{Z}, h)$ be defined by Eq. (14). Then, $f_{\vec{g}}$ is $O_{\vec{g}}(3)$-equivariant. More importantly, For any $O_{\vec{g}}(3)$-equivariant function $\hat{f}(\vec{Z}, h)$, there always exists an MLP $\sigma$ satisfying $\|\hat{f} - f_{\vec{g}}\| < \epsilon$ for arbitrarily small positive value $\epsilon$.*

The proof is similar to Proposition 1 but with certain extensions. We first derive the following three lemmas.

**Lemma 3.** *For any $O_{\vec{g}}(3)$-equivariant function $\hat{f}(\vec{Z}, h)$, it must fall into the subspace spanned by the columns of $[\vec{Z}, \vec{g}]$, namely, there exists a function $s(\vec{Z}, h)$, satisfying $\hat{f}(\vec{Z}, h) = [\vec{Z}, \vec{g}]s(\vec{Z}, h)$.*

*Proof.* The proof is similar to Lemma 3. Suppose $\vec{Z}^{\perp}$ is the orthogonal complement of the column space of $[\vec{Z}, \vec{g}]$. Then there must exit functions $s(\vec{Z}, h)$ and $s^{\perp}(\vec{Z}, h)$, satisfying $\hat{f}(\vec{Z}, h) = [\vec{Z}, \vec{g}]s(\vec{Z}, h) + \vec{Z}^{\perp}s^{\perp}(\vec{Z}, h)$. We can always find an orthogonal transformation $O \in O_{\vec{g}}$ allowing $O\vec{Z} = \vec{Z}$, $O\vec{g} = \vec{g}$ while $O\vec{Z}^{\perp} = -\vec{Z}^{\perp}$. With this transformation $O$, we have $\hat{f}(O\vec{Z}, h) = \hat{f}(\vec{Z}, h) = [\vec{Z}, \vec{g}]s(\vec{Z}, h) + \vec{Z}^{\perp}s^{\perp}(\vec{Z}, h)$, and $O\hat{f}(\vec{Z}, h) = [\vec{Z}, \vec{g}]s(\vec{Z}, h) - \vec{Z}^{\perp}s^{\perp}(\vec{Z}, h)$. The equivariance property of $\hat{f}$ implies $[\vec{Z}, \vec{g}]s(\vec{Z}, h) + \vec{Z}^{\perp}s^{\perp}(\vec{Z}, h) = [\vec{Z}, \vec{g}]s(\vec{Z}, h) - \vec{Z}^{\perp}s^{\perp}(\vec{Z}, h)$, which derives $s^{\perp}(\vec{Z}, h) = 0$. Hence, the proof is concluded. $\square$

**Lemma 4.** *If the $O_{\vec{g}}(3)$-equivariant function $\hat{f}(\vec{Z}, h)$ lies in the subspace spanned by the columns of $[\vec{Z}, \vec{g}]$, then there exists a $O_{\vec{g}}(3)$-invariant function $\eta$ satisfying $\hat{f}(\vec{Z}, h) = [\vec{Z}, \vec{g}]\eta(\vec{Z}, h)$.*

*Proof.* We assume the rank of $[\vec{Z}, \vec{g}]$ is $r$ ($r \leq \min\{3, m+1\}$). By performing the compact SVD decomposition on $[\vec{Z}, \vec{g}]$, we devise $[\vec{Z}, \vec{g}] = US_rV^{\top}$, where $S_r \in \mathbb{R}^{r\times r}$ is a square diagonal matrix with positive diagonal elements, $U \in \mathbb{R}^{3\times r}$, $V \in \mathbb{R}^{(m+1)\times r}$, and $U^{\top}U = V^{\top}V = I_r$.

Since $\hat{f}(\vec{Z}, h)$ lies in the subspace spanned by the columns of $[\vec{Z}, \vec{g}]$, there exists a function $s(\vec{Z}, h)$ allowing $\hat{f}(\vec{Z}, h) = [\vec{Z}, \vec{g}]s(\vec{Z}, h)$. With applying the SVD decomposition, we have

$$\hat{f}(\vec{Z}, h) = US_rV^{\top}s(\vec{Z}, h). \tag{15}$$

Given that $\hat{f}$ is $O_{\vec{g}}(3)$-equivariant, it means $\hat{f}(O\vec{Z}, h) = O\hat{f}(\vec{Z}, h)$. By substituting this equation into Eq. (15),

$$OUS_rV^{\top}s(O\vec{Z}, h) = OUS_rV^{\top}s(\vec{Z}, h)$$
$$\Leftrightarrow V^{\top}s(O\vec{Z}, h) = V^{\top}s(\vec{Z}, h). \tag{16}$$

As $V$ is $O_{\vec{g}}(3)$-invariant, $V^{\top}s(\vec{Z}, h)$ is an $O_{\vec{g}}(3)$-invariant function. Define $\eta(\vec{Z}, h) := VV^{\top}s(\vec{Z}, h)$, which is apparently $O_{\vec{g}}(3)$-invariant. Then we have $\hat{f}(\vec{Z}, h) = US_rV^{\top}s(\vec{Z}, h) = US_rV^{\top}VV^{\top}s(\vec{Z}, h) = [\vec{Z}, \vec{g}]\eta(\vec{Z}, h)$.

$\square$

**Lemma 5.** *If the function $\eta(\vec{Z}, h)$ is $O_{\vec{g}}(3)$-invariant, it is of the form $\eta(\vec{Z}, h) = \sigma([\vec{Z}, \vec{g}]^{\top}[\vec{Z}, \vec{g}], h)$ for a certain function $\sigma$.*

*Proof.* For any two inputs $\vec{Z}_1, \vec{Z}_2$, we claim that

$$\exists O \in O_{\vec{g}}(3), \vec{Z}_1 = O\vec{Z}_2 \Leftrightarrow [\vec{Z}_1, \vec{g}]^{\top}[\vec{Z}_1, \vec{g}] = [\vec{Z}_2, \vec{g}]^{\top}[\vec{Z}_2, \vec{g}]. \tag{17}$$

The sufficiency direction $\Rightarrow$ is obvious. We only need to prove the necessity $\Leftarrow$.

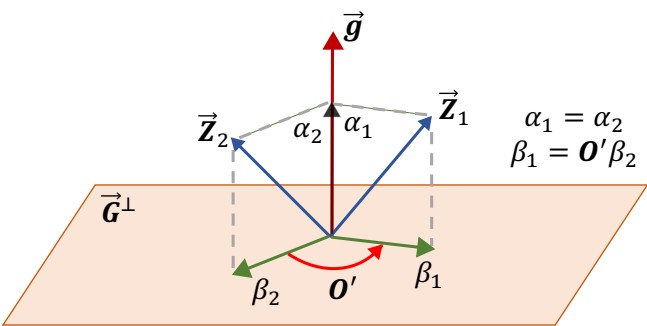

Figure 7: A geometric illustration of the proof for Lemma 5.

We denote by $\vec{G}^\perp \in \mathbb{R}^{3\times 2}$ the orthogonal complement of $\vec{g}$. Then, we have the decompositions:

$$\vec{Z}_1 = \vec{g}\alpha_1 + \vec{G}^\perp\beta_1; \tag{18}$$

$$\vec{Z}_2 = \vec{g}\alpha_2 + \vec{G}^\perp\beta_2; \tag{19}$$

where $\alpha_1, \alpha_2 \in \mathbb{R}^{1\times m}$, $\beta_1, \beta_2 \in \mathbb{R}^{2\times m}$. The RHS of Eq. (17) indicates $\vec{g}^\top\vec{Z}_1 = \vec{g}^\top\vec{Z}_2$, and $\vec{Z}_1^\top\vec{Z}_1 = \vec{Z}_2^\top\vec{Z}_2$, hence we have $\alpha_1 = \alpha_2$, and

$$\begin{aligned}
\beta_1^\top\beta_1 &= (\vec{G}^\perp\beta_1)^\top(\vec{G}^\perp\beta_1) \\
&= (\vec{Z}_1 - \vec{g}\alpha_1)^\top(\vec{Z}_1 - \vec{g}\alpha_1) \\
&= \vec{Z}_1^\top\vec{Z}_1 - \alpha_1^\top\alpha_1 \\
&= \vec{Z}_2^\top\vec{Z}_2 - \alpha_2^\top\alpha_2 \\
&= \beta_2^\top\beta_2.
\end{aligned} \tag{20}$$

According to Lemma 1 in [5], $\beta_1^\top\beta_1 = \beta_2^\top\beta_2 \Leftrightarrow \exists O' \in O(2), \beta_1 = O'\beta_2$.

Now we define $O = \vec{g}\vec{g}^\top + \vec{G}^\perp O'(\vec{G}^\perp)^\top$. Clearly, $O^\top O = \vec{g}\vec{g}^\top + \vec{G}^\perp(\vec{G}^\perp)^\top = I_3$, and $O^\top\vec{g} = \vec{g}$, which means $O \in O_{\vec{g}}(3)$. More interestingly,

$$\begin{aligned}
O\vec{Z}_2 &= \vec{g}\vec{g}^\top\vec{Z}_2 + \vec{G}^\perp O'(\vec{G}^\perp)^\top\vec{Z}_2 \\
&= \vec{g}\alpha_2 + \vec{G}^\perp O'\beta_2 \\
&= \vec{g}\alpha_1 + \vec{G}^\perp\beta_1 \\
&= \vec{Z}_1,
\end{aligned} \tag{21}$$

which concludes the proof of the claim. Fig. 7 provides a geometrical explanation of our proof above, where the transformation $O'$ is actually the projection of $O$ onto the subspace $\vec{G}^\perp$.

For any $O_{\vec{g}}(3)$-invariant function $\eta(\vec{Z}, h)$, it is a function of the equivalent class, that is, $\eta(\vec{Z}, h) = \sigma(\{\vec{Z}\}, h)$, where the equivalent class is defined as $\{\vec{Z}\} := \{O\vec{Z} \mid O \in O_{\vec{g}}(3)\}$. The above claim in (17) states that such equivalent class $\{\vec{Z}\}$ maps to $[\vec{Z}, \vec{g}]^\top[\vec{Z}, \vec{g}]$ in a one-to-one way. Therefore, we must arrive at $\eta(\vec{Z}, h) = \sigma([\vec{Z}, \vec{g}]^\top[\vec{Z}, \vec{g}], h)$.

$\square$

By making use of Lemma 3-5, we immediately obtain $f_{\vec{g}}(\vec{Z}, h) = [\vec{Z}, \vec{g}]\sigma([\vec{Z}, \vec{g}]^\top[\vec{Z}, \vec{g}], h)$. In accordance with the universality of MLP [3, 4], we can always find an MLP that approximates $\sigma$ up to an accuracy $\frac{\epsilon}{G}$, where $G$ is the bound of $\|[\vec{Z}, \vec{g}]\|$, which implies $\|\hat{f} - f_{\vec{g}}\| < \|[\vec{Z}, \vec{g}]\|\frac{\epsilon}{G} < \epsilon$, where $f_{\vec{g}}$ is defined in Eq. (14). The proof of Theorem 1 is finished.

Note that $f$ by Eq. (14) can also be considered as a function of both $\vec{Z}$ and $\vec{g}$, and it is universal according to Proposition 1. When $f$ reduces to a function of $\vec{Z}$ by fixing $\vec{g}$, then by Theorem 1, it is

still universal with respect to the subgroup that leaves $\vec{g}$ unchanged. This phenomenon shows the universality holds universally no matter which input variable of Eq. (14) we fix.

Although this paper mainly focuses on the 3-dimension group O(3), our theorems and proofs above are generalizable to the $d$-dimension group O($d$), when $d$ is not restricted to 3 and the external field that breaks the symmetry has more than 1 direction, namely the gravity vector $\vec{g} \in \mathbb{R}^{3 \times 1}$ becomes a directional matrix $\vec{G} \in \mathbb{R}^{d \times d'} (d' < d)$.

## A.2 Proof of Theorem 2

**Theorem 2.** *The message passing $\varphi$ (Eq. 4) is $O_{\vec{g}}(3)$-equivariant.*

*Proof.* We prove step by step in the specifications of Eq. (4) from Eq. (5)-(8). For better clarity, we denote the variables after applying the transformation $O \in O_{\vec{g}}(3)$ with the superscript $*$.

For any $O \in O_{\vec{g}}(3)$, we immediately have $(\vec{Z}_i^*, h_i^*) = (O\vec{Z}_i, h_i)$, and

$$(\vec{C}_k^*, c_k^*) = \left( \frac{1}{|\{i : o(i) = k\}|} \sum_{\{i:o(i)=k\}} \vec{Z}_i^*, \sum_{\{i:o(i)=k\}} h_i^* \right),$$

$$= \left( \frac{1}{|\{i : o(i) = k\}|} \sum_{\{i:o(i)=k\}} O\vec{Z}_i, \sum_{\{i:o(i)=k\}} h_i \right),$$

$$= (O\vec{C}_k, c_k).$$

Therefore, for Eq. (5),

$$\vec{Z}_{ij}^* = (\vec{Z}_i^* \ominus \vec{C}_{o(i)}^*) \| (\vec{Z}_j^* \ominus \vec{C}_{o(j)}^*) \| (\vec{Z}_i^* \ominus \vec{Z}_j^*),$$

$$= (O\vec{Z}_i \ominus O\vec{C}_{o(i)}) \| (O\vec{Z}_j \ominus O\vec{C}_{o(j)}) \| (O\vec{Z}_i \ominus O\vec{Z}_j),$$

$$= O\left( (\vec{Z}_i \ominus \vec{C}_{o(i)}) \| (\vec{Z}_j \ominus \vec{C}_{o(j)}) \| (\vec{Z}_i \ominus \vec{Z}_j) \right) = O\vec{Z}_{ij},$$

which is $O_{\vec{g}}(3)$-equivariant. Similarly for $h_{ij}$ in Eq. (6), it is invariant, *i.e.*, $h_{ij}^* = h_{ij}$. Since $\phi_{\vec{g}}$ in Eq. (7) is designed to be subequivariant ($O_{\vec{g}}(3)$-equivariant), by definition we immediately derive that $(\vec{M}_{ij}^*, m_{ij}^*) = \phi_{\vec{g}}\left( \vec{Z}_{ij}^*, h_{ij}^* \right) = \phi_{\vec{g}}\left( O\vec{Z}_{ij}, h_{ij} \right) = (O\vec{M}_{ij}, m_{ij})$.

Finally, for the aggregation and update in Eq. (8), it is derived as

$$(\vec{Z}_i^{\prime*}, h_i^{\prime*}) = (\vec{Z}_i^*, h_i^*) + \psi_{\vec{g}}\left( (\sum_{j \in \mathcal{N}(i)} \vec{M}_{ij}^*) \| (\vec{Z}_i^* \ominus \vec{C}_{o(i)}^*), (\sum_{j \in \mathcal{N}(i)} m_{ij}^*) \| h_i^* \| c_{o(i)}^* \right),$$

$$= (O\vec{Z}_i, h_i) + \psi_{\vec{g}}\left( (\sum_{j \in \mathcal{N}(i)} O\vec{M}_{ij}) \| (O\vec{Z}_i \ominus O\vec{C}_{o(i)}), (\sum_{j \in \mathcal{N}(i)} m_{ij}) \| h_i \| c_{o(i)} \right),$$

$$= (O\vec{Z}_i, h_i) + \psi_{\vec{g}}\left( O((\sum_{j \in \mathcal{N}(i)} \vec{M}_{ij}) \| (\vec{Z}_i \ominus \vec{C}_{o(i)})), (\sum_{j \in \mathcal{N}(i)} m_{ij}) \| h_i \| c_{o(i)} \right),$$

$$= (O\vec{Z}_i', h_i'),$$

which concludes the proof by showing that $\vec{Z}_i'$ is $O_{\vec{g}}(3)$-equivariant and $h_i'$ is $O_{\vec{g}}(3)$-invariant. $\square$

Indeed, by leveraging Theorem 2, it is also straightforward that the resulting SGNN is also $O_{\vec{g}}(3)$-equivariant, since its components $\varphi_1$, $\varphi_2$, and $\varphi_3$ are all $O_{\vec{g}}(3)$-equivariant functions.

**Remark on translation equivariance.** Regarding translation equivariance, the operation "$\ominus$" always results in translation-invariant representations. Therefore, $\vec{Z}_{ij}$ is translation invariant, and so does $h_{ij}$. Following this induction, the intermediate results until the output of $\psi_{\vec{g}}$ are all translation-invariant. By adding $(\vec{Z}_i, h_i)$ to the final output, it is clear to see that $\vec{Z}_i'$ is translation-equivariant and $h_i'$ is translation-invariant.

## A.3  Theoretical Comparisons Between EGNN, GMN, and SGNN

In this sub-section, we theoretically reveal that both EGNN and GMN are special cases of SGNN by choosing specific forms of MLP in $\phi_{\vec{g}}$ of Eq. (7) and $\psi_{\vec{g}}$ of Eq. (8). We provide an illustration from the architectural view in Fig. 8.

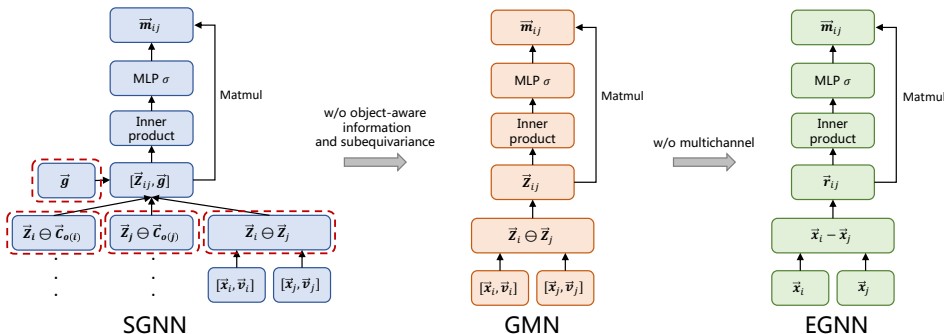

Figure 8: An illustration of the comparison between SGNN, GMN, and EGNN.

We first prove that SGNN can reduce to GMN by choosing specific form of the MLP $\sigma$. Intuitively, this reduction can be realized by masking the channels related to $\vec{g}$ and the object-aware information $\vec{C}_{o(i)}$ and $\vec{C}_{o(j)}$ from the input and the corresponding output before taking the matrix multiplication. We summarize this into the following lemma.

**Lemma 6.** *Consider arbitrary 3D multichannel vectors $\vec{Z}_1 \in \mathbb{R}^{3 \times m_1}$ and $\vec{Z}_2 \in \mathbb{R}^{3 \times m_2}$. Let $f_{\sigma_1} = [\vec{Z}_1, \vec{Z}_2] \sigma_1 \left( [\vec{Z}_1, \vec{Z}_2]^\top [\vec{Z}_1, \vec{Z}_2] \right), g_{\sigma_2} = \vec{Z}_1 \sigma_2 \left( \vec{Z}_1^\top \vec{Z}_1 \right).$ Then, for any $\sigma_2$, there exists $\sigma_1$, s.t. $\|f_{\sigma_1} - g_{\sigma_2}\| < \epsilon$ for arbitrary small positive value $\epsilon$.*

*Proof.* The input of $\sigma_1$ can be rewritten as $\begin{bmatrix} \vec{Z}_1^\top \vec{Z}_1 & \vec{Z}_1^\top \vec{Z}_2 \\ \vec{Z}_2^\top \vec{Z}_1 & \vec{Z}_2^\top \vec{Z}_2 \end{bmatrix}$. Denote $f_{\text{in}}(X) = \begin{bmatrix} 1 & 0 \\ 0 & 0 \end{bmatrix} X$ and $f_{\text{out}}(X) = \begin{bmatrix} X \\ 0 \end{bmatrix}$. Then, when choosing $\sigma_1 = f_{\text{out}} \circ \sigma_2 \circ f_{\text{in}}$, we have

$$
\begin{aligned}
f_{\sigma_1} &= [\vec{Z}_1, \vec{Z}_2] \sigma_1 \left( [\vec{Z}_1, \vec{Z}_2]^\top [\vec{Z}_1, \vec{Z}_2] \right), \\
&= [\vec{Z}_1, \vec{Z}_2] f_{\text{out}} \circ \sigma_2 \circ f_{\text{in}} \left( \begin{bmatrix} \vec{Z}_1^\top \vec{Z}_1 & \vec{Z}_1^\top \vec{Z}_2 \\ \vec{Z}_2^\top \vec{Z}_1 & \vec{Z}_2^\top \vec{Z}_2 \end{bmatrix} \right), \\
&= [\vec{Z}_1, \vec{Z}_2] f_{\text{out}} \circ \sigma_2 \left( \vec{Z}_1^\top \vec{Z}_1 \right), \\
&= [\vec{Z}_1, \vec{Z}_2] \begin{bmatrix} \sigma_2 \left( \vec{Z}_1^\top \vec{Z}_1 \right) \\ 0 \end{bmatrix}, \\
&= \vec{Z}_1 \sigma_2 \left( \vec{Z}_1^\top \vec{Z}_1 \right) = g_{\sigma_2}.
\end{aligned}
$$

Therefore, by the universal approximation of MLP [3, 4], we know that for any $g_{\sigma_2}$ parameterized by $\sigma_2$, there exists $\sigma_1 = f_{\text{out}} \circ \sigma_2 \circ f_{\text{in}}$ that can approximate $g_{\sigma_2}$ with $f_{\sigma_1}$ by arbitrarily small error $\epsilon$.  □

Leveraging this lemma directly gives the following theorem.

**Theorem 3.** *Let the symbol $f_{\sigma_1} \succeq g_{\sigma_2}$ denotes that for any $g_{\sigma_2}$ parameterized by $\sigma_2$, there exists $\sigma_1$ satisfying $\|f_{\sigma_1} - g_{\sigma_2}\| < \epsilon$ for arbitrarily small positive value $\epsilon$. Then, SOMP $\succeq$ GMN $\succeq$ EGNN.*

*Proof.* **1. SOMP $\succeq$ GMN.** Let $\vec{Z}_1 = [\vec{Z}_i \ominus \vec{Z}_j]$, and $\vec{Z}_2 = [\vec{Z}_i \ominus \vec{C}_{o(i)}, \vec{Z}_j \ominus \vec{C}_{o(j)}, \vec{g}]$. Using Lemma 6 immediately shows that SOMP $\succeq$ GMN. **2. GMN $\succeq$ EGNN.** Similarly, we choose

$\vec{Z}_1 = [\vec{x}_i - \vec{x}_j]$, and $\vec{Z}_2 = [\vec{v}_i, \vec{v}_j]$, and by Lemma 6 we have GMN $\succeq$ EGNN, which concludes the proof. $\qquad\square$

By these theoretical derivations we are able to show that SOMP indeed has stronger expressivity than GMN and EGNN, by leveraging object-aware information as well as the subequivariance depicted by vector $\vec{g}$.

# B  Implementation Details

## B.1  Hyper-parameters and Training Details

We utilize the codebase provided by Physion [1] for particle-based methods[1]. This repository contains the implementation of GNS [9] and DPI [7]. For DPI, we notice that it has been optimized in the RigidFall task by [6][2], and we thus adopt their optimized version on RigidFall. As for EGNN [10] and GMN [5], we resort to their original implementations[3][4], respectively. The datasets and code repositories are released under MIT license.

We basically follow the hyper-parameters suggested by Physion. In detail, for GNS and DPI, we use a hidden dimension of 200 for the node update function $\psi$ and 300 for the message computation function $\phi$, each of which consists of 3 layers with ReLU as the activation function. The iteration step is set to 10 for GNS and 2 for DPI due to its multi-stage hierarchical modeling. We use an Adam optimizer with initial learning rate 0.0001, betas (0.9, 0.999), and a Plateau scheduler with a patience of 3 epochs and decaying factor 0.8. For EGNN, GMN, and SGNN, we still build upon the above hyper-parameters with very minor modifications. We adopt hidden dimension 200 uniformly for $\psi$ and $\phi$ with SiLU activation function, and 4 iterations in $\varphi_1$, $\varphi_2$, and $\varphi_3$ for SGNN, while 10 for EGNN and GMN. We use an early-stopping of 10 epochs. We use a batch size of 1 on Physion due to the large size of each system and 8 on RigidFall due to its relatively small size. Besides, we also inject noise during training for better test-time long-term rollout prediction, exactly following the settings of Physion and RigidFall [6]. The scale is set to 3e-4 in Physion and 0.05 of the std in RigidFall. The cutoff radius $\gamma$ is set to be 0.08 on both datasets. On both datasets, we only use the state information of last frame $t$ as input to predict the information of frame $t + 1$. The experiments are conducted on single card NVIDIA Tesla V100 GPU.

Notably, for EGNN-S and GMN-S, we make a modification to their updates of the velocity, namely,

$$\vec{\mathbf{v}}_i^{l+1} = \phi_v(\mathbf{h}_i^l)\vec{\mathbf{v}}_i^l + \underline{\phi_g(\mathbf{h}_i^l)\vec{\mathbf{g}}} + \sum_{j \in \mathcal{N}(i)} (\vec{\mathbf{x}}_i - \vec{\mathbf{x}}_j)\phi_x(\mathbf{m}_{ij}),$$

where the underlined term highlights our adaptation, $\phi_v, \phi_g, \phi_x$ are all MLPs, and the superscript $l$ indicates the iteration step. The intuition of this term is similar to adding a gravitational acceleration term to the update of velocity. This formulation meets $O_{\vec{g}}(3)$-equivariance.

As for the data splits, we strictly follow Physion and RigidFall. In detail, for Physion, the full training set contains 2000 trajectories in each scenario, which is then split into training and validation with the ratio 9:1. The testing set contains 200 trajectories. For RigidFall, the full training set contains 5000 trajectories, which is also split into training and validation with ratio 9:1. Particularly, to study the data-efficiency of different models, we sub-sample multiple training sets with sizes 200, 500, 1000, 5000, as illustrated in Table 3 in the paper.

Besides, in the implementation we also employ a normalization before feeding the inner product into the MLP, *i.e.*, we normalize the inner product by $\vec{Z}^\top \vec{Z}/\|\vec{Z}^\top \vec{Z}\|_F$ in Eq. (3), as suggested in GMN [5] to control the expanding variance in scale brought by the inner product for better numerical stability. This is similarly adopted in Eq. (9) for $[\vec{Z}, \vec{g}]$, where we also propose to dynamically control the scale of $\vec{g}$ by $\eta(\boldsymbol{h}) \in \mathbb{R}$ where $\eta$ is a lightweight MLP. We find these considerations generally leads to faster convergence.

---

[1] https://github.com/htung0101/Physion-particles
[2] https://github.com/YunzhuLi/VGPL-Dynamics-Prior
[3] https://github.com/vgsatorras/egnn
[4] https://github.com/hanjq17/GMN

### B.2 Computational Complexity

In this sub-section, we compare the computational budget of SGNN to those of the baselines, aiming to illustrate that the superior performance brought by SGNN stems from our design of subequivariance and hierarchical modeling, but not more computational overhead or more parameters used.

Generally, the computational complexity of the models is approached by $O(KT|\mathcal{E}|)$, where $K$ is the number of stages, $T$ is the number of message passing steps in each stage, and $|\mathcal{E}|$ measures the number of edges in the interaction graph. Among all the models, they can be characterized into non-hierarchical methods including GNS, EGNN, and GMN, as well as hierarchical methods including DPI and our SGNN. For the non-hierarchical models, it has been observed in [9] that generally a larger number of propagation iterations $T$ is required for better performance, which is set to 10 in the implementation. For the hierarchical methods, it does not require such a large number, which is set to 2 for DPI following their original setup and 4 in SGNN. By this means, we have carefully controlled the computational budget by making the total number of message passing iterations nearly the same for all methods, since DPI requires in total $K = 4$ stages (leaf-leaf, leaf-root, root-root, root-leaf) with each stage involving 2 steps, and SGNN requires $K = 3$ stages $\varphi_1$, $\varphi_2$, and $\varphi_3$, with each stage having 4 steps. GNS, EGNN, and GMN only have one stage, but need 10 steps in this stage. Besides, it is also worth noticing that SGNN employs edge separation, which further brings down the cost when computing message passing with $|\mathcal{E}_{\text{inter}}|, |\mathcal{E}_{\text{inner}}|, |\mathcal{E}_{\text{obj}}| < |\mathcal{E}|$ edges. Regarding the size of the networks, we reuse the hyper-parameters, *e.g.*, the number of layers in MLP and the hidden dimension, of the baselines for SGNN, which makes SGNN nearly as the same size as DPI.

To further illustrate, we provide the total number of parameters and the average training time per step in seconds on Physion Dominoes in Table 4, which indicates that SGNN has a moderate number of parameters and still enjoys fast training speed.

Table 4: Number of parameters (#Param) and average training time per step on Physion Dominoes.

|                | GNS [9]       | DPI [7]       | EGNN [10]     | GMN [5]       | SGNN          |
| -------------- | ------------- | ------------- | ------------- | ------------- | ------------- |
| #Param         | 0.54M         | 1.98M         | 0.45M         | 0.51M         | 1.50M         |
| Time (seconds) | $0.40 \pm 0.02$ | $0.35 \pm 0.03$ | $0.08 \pm 0.01$ | $0.09 \pm 0.01$ | $0.11 \pm 0.02$ |

## C   More Experiment Results

### C.1   Motivating Example

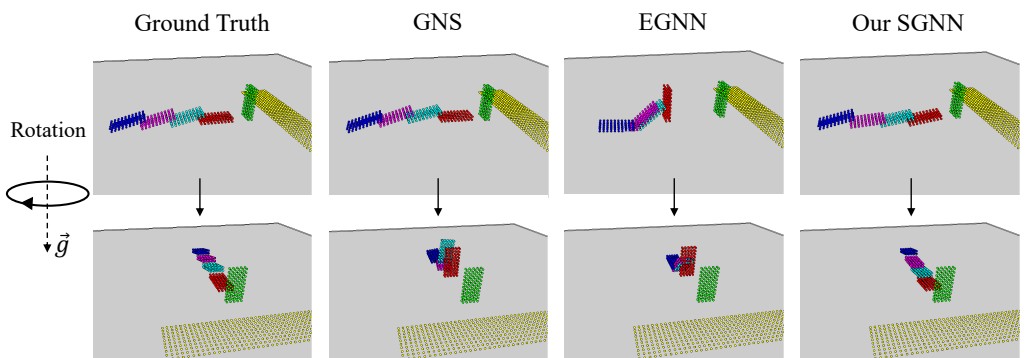

Figure 9: A motivating example on Physion Dominoes. GNS is able to produce accurate prediction only in the same direction as the training trajectories. EGNN fails to learn the complex interactions with gravity involved. Our model predicts very accurately regardless of any valid rotations applied.

We provide a motivating example in Fig. 9. We have the following observations: **1.** Physical laws abide by symmetry. The dynamics of the dominoes falling from left to right will exactly be preserved the same way if we apply a rotation along the gravitational axis $\vec{g}$; **2.** Without any guarantee of

equivariance, models like GNS fail to preserve such symmetry. For example, if all dominoes in the training data are falling from left to right, GNS might be able to produce accurate prediction if the testing trajectory is also aligned from left to right, but will perform poorly if a rotation is adopted; **3.** Equivariant models like EGNN and GMN are able to preserve the symmetry, *i.e.*, their prediction will rotate/translate together with the input. However, they are enforcing E(3)-equivariance, which is too strong that limits the expressivity of the model when the equivariance is violated by the existence of gravity; **4.** Our model SGNN takes into consideration the desirable subequivariance as well as object-aware message passing, yielding very accurate prediction while being invulnerable to any test-time rotation along the vertical axis.

## C.2 Rollout MSE on Physion

Table 5: Rollout MSE on Physion when $t = 10$.

|  | Dominoes | Contain | Link | Drape | Support | Drop | Collide | Roll |
|---|---|---|---|---|---|---|---|---|
| GNS* [9] | 0.19 | 3.76 | 9.89 | 18.51 | 4.12 | 2.59 | 1.58 | 0.60 |
| DPI* [7] | 0.15 | 2.41 | 6.56 | 18.43 | **3.67** | 2.39 | 1.66 | 0.63 |
| GNS [9] | 1.01 | 4.25 | 16.23 | 23.72 | 4.20 | 2.57 | 5.35 | 0.80 |
| DPI [7] | 1.04 | 3.13 | 12.88 | 35.05 | 4.37 | 1.9 | 8.02 | 2.05 |
| GNS-Rot [9] | 0.27 | 3.95 | 10.39 | 29.85 | 4.69 | 2.04 | 1.53 | 0.63 |
| DPI-Rot [7] | 0.22 | **2.27** | 6.37 | 26.94 | 4.61 | 1.88 | 1.64 | 1.41 |
| EGNN [10] | 0.31 | 5.91 | 12.93 | 39.81 | 5.25 | 1.86 | 4.22 | 1.01 |
| GMN [5] | 0.59 | 8.88 | 19.36 | 39.70 | 9.08 | 3.16 | 6.13 | 2.03 |
| SGNN | **0.09** | 2.32 | **4.98** | **17.23** | 4.52 | **1.37** | **1.34** | **0.53** |

Table 6: Rollout MSE ($\times 10^1$) on Physion when $t = 35$.

|  | Dominoes | Contain | Link | Drape | Support | Drop | Collide | Roll |
|---|---|---|---|---|---|---|---|---|
| GNS* [9] | 0.16 | 4.83 | 10.95 | 9.04 | **9.59** | 4.78 | 3.28 | 0.39 |
| DPI* [7] | 0.14 | 2.36 | 9.49 | 32.97 | 28.97 | 1.73 | 4.07 | 0.38 |
| GNS [9] | 0.53 | 5.28 | 16.56 | 9.53 | 9.74 | 5.92 | 8.10 | 0.43 |
| DPI [7] | 0.56 | 3.39 | 16.29 | 30.21 | 17.07 | 0.98 | 11.40 | 1.54 |
| GNS-Rot [9] | 0.22 | 4.32 | 15.54 | 12.13 | 9.80 | 1.79 | 3.02 | 0.37 |
| DPI-Rot [7] | 0.35 | 4.65 | 11.04 | 12.02 | 53.69 | 1.24 | 3.68 | 1.54 |
| EGNN [10] | 0.30 | 6.36 | 19.99 | 13.41 | 14.81 | 0.96 | 5.96 | 0.66 |
| GMN [5] | 0.39 | 7.25 | 22.97 | 12.33 | 16.60 | 1.53 | 7.93 | 0.84 |
| SGNN | **0.07** | **2.16** | **7.01** | **8.14** | 13.55 | **0.70** | **2.80** | **0.31** |

We provide the rollout MSE when $t = 10$ and $t = 35$ on Physion in Table 5 and 6, respectively. Our SGNN gives the best results in 7 out of all 8 scenarios, especially favorable on long-term prediction.

## C.3 Learning curves

We provide the learning curves including training loss and validation loss on RigidFall in Fig. 10 with training data size 200 and 500, respectively. Our core observations here include **1.** The non-equivariant model DPI tends to suffer from *overfitting*, since it lacks the inductive bias of symmetry. This can be observed from the curve that its training loss can reach a very low level (*e.g.*, better than SGNN when |Train| = 200), but the validation is not promising. There is generally a big gap between training and validation. Without the equivariance constraint, it may overfit the directions only existing in the training data and fail to generalize to validation set; **2.** The E(3)-equivariant model EGNN may underfit the training data, since it is restricted by an over-strong constraint that indeed fails to capture the real dynamics in the training data, though its generalization is desirable with a very small gap between training and validation; **3.** Our SGNN, by leveraging subequivariance, fits the training data

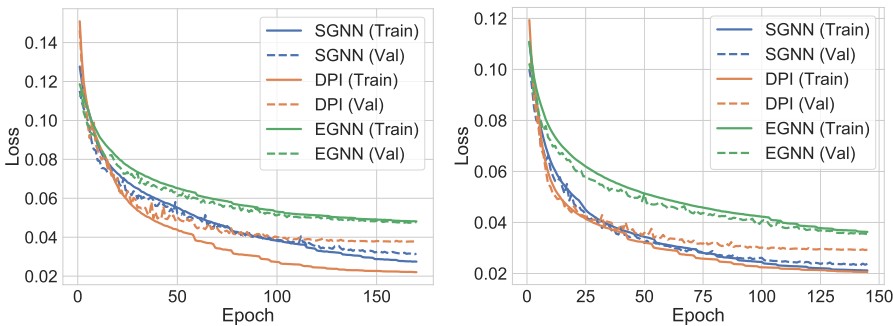

Figure 10: Learning curve comparisons on RigidFall. Left: $|\text{Train}| = 200$; Right: $|\text{Train}| = 500$.

well while yields strong generalization, achieving the lowest validation loss. These observations from the learning curves well align with our analyses and verify the efficacy of our design.

### C.4 Experiment on Hamiltonian-based NNs

We augment SGNN and EGNN by a Hamiltonian integrator. Details of the implementation include:

- We leverage a sum-pooling over the output scalar feature ($\mathbf{h}_i$) as the Hamiltonian of the system, i.e., $\mathcal{H} \in \mathbb{R} = \sum_{i=1}^{N} \mathbf{h}_i$.
- We employ a RK1 integrator to conduct Hamiltonian update, i.e., $(\dot{\vec{\mathbf{q}}}, \dot{\vec{\mathbf{p}}}) = (\frac{\partial \mathcal{H}}{\partial \vec{\mathbf{p}}}, -\frac{\partial \mathcal{H}}{\partial \vec{\mathbf{q}}})$.

One thing worth noticing here is that we are assuming the particles possess uniform mass, so that $\vec{\mathbf{q}}, \vec{\mathbf{p}}$ can be derived from $\vec{\mathbf{x}}, \vec{\mathbf{v}}$, respectively. We name the variants as SGNN-H and EGNN-H ("H" stands for Hamiltonian), and evaluate them on Physion. The results are displayed in Table 7.

Table 7: Comparison with the Hamiltonian-based variants.

|  | Domino | Contain | Link | Drape | Support | Drop | Collide | Roll |
|---|---|---|---|---|---|---|---|---|
| EGNN [10] | 61.3 | 66.0 | 52.7 | 54.7 | 60.0 | 63.3 | 76.7 | 79.8 |
| EGNN-H [10, 8] | 52.0 | 58.1 | 54.0 | 54.3 | 51.1 | 54.7 | 75.7 | 75.3 |
| SGNN | 89.1 | 78.1 | 73.3 | 60.6 | 71.2 | 74.3 | 85.3 | 84.2 |
| SGNN-H | 69.9 | 66.0 | 61.1 | 60.3 | 55.3 | 62.0 | 79.3 | 78.7 |

Table 8: Average training time per step (in seconds) on Physion Dominoes.

| EGNN [10] | EGNN-H [10, 8] | SGNN | SGNN-H |
|---|---|---|---|
| 0.08±0.01 | 0.44±0.02 | 0.11±0.02 | 0.48±0.03 |

Adding Hamiltonian into EGNN and SGNN generally leads to detrimental performance. We speculate that it is probably due to the dissipative forces as well as highly complex interactions in Physion. Moreoever, as illustrated in Table 8, the Hamiltonian module brings significant computation overhead during training. This result suggests that it may not be beneficial to involve such strong physical inductive bias for the scenarios in Physion.

### C.5 Experiment on Steerable SE(2) GNN

We also implement a baseline that leverages the idea in [12, 2] but extends from CNN to GNN. Indeed, [12, 2] are steerable CNNs, and these works have not offered available implementations on GNNs. We have tried our best to compare this idea with our model. Specifically, we implement "Steerable-SE(2)-GNN", that iterates the message passing as specified below. Consider the message computation for the edge $e_{ij} \in \mathcal{E}$ connecting node $i$ and $j$.

- Compute the translation-invariant radial vector: $\vec{\mathbf{x}}_{ij} = \vec{\mathbf{x}}_i - \vec{\mathbf{x}}_j$.
- Project $\vec{\mathbf{x}}_{ij}$ onto $\vec{\mathbf{g}}$: $v \in \mathbb{R} = \frac{\vec{\mathbf{x}}_{ij} \cdot \vec{\mathbf{g}}}{\|\vec{\mathbf{g}}\|}$, and $\vec{\mathbf{u}} \in \mathbb{R}^2 = ((\vec{\mathbf{x}}_{ij} - v\vec{\mathbf{g}}) \cdot \vec{\mathbf{m}}, ((\vec{\mathbf{x}}_{ij} - v\vec{\mathbf{g}}) \cdot \vec{\mathbf{n}})$, where $\vec{\mathbf{m}}, \vec{\mathbf{n}}$ are two orthonormal bases vertical to $\vec{\mathbf{g}}$.
- Derive the type-0 message as $\mathbf{m}_{ij} = \mathrm{MLP}_1(\sum_l w_l^{01} k_l^{01}(\vec{\mathbf{u}}) \cdot \vec{\mathbf{u}}, v, \|\vec{\mathbf{u}}\|, \mathbf{h}_i, \mathbf{h}_j)$.
- Derive the type-1 message as $\vec{\mathbf{M}}_{ij} = (\sum_l w_l^{10} k_l^{10}(\vec{\mathbf{u}})\mathbf{m}_{ij} + \sum_l w_l^{11} k_l^{11}(\vec{\mathbf{u}}) \cdot \vec{\mathbf{u}}) \cdot \mathrm{MLP}_2(\mathbf{m}_{ij})$.
- Aggregate and update type-0 feature: $\mathbf{h}_i' = \mathrm{MLP}_3(\sum_{j \in \mathcal{N}(i)} \mathbf{m}_{ij}, \mathbf{h}_i)$.
- Aggregate and update type-1 feature: $\vec{\mathbf{M}}_i = \sum_{j \in \mathcal{N}(i)} \vec{\mathbf{M}}_{ij}$, $\vec{\mathbf{x}}_i' = \vec{\mathbf{x}}_i + \mathrm{MLP}_4(\|\vec{\mathbf{M}}_i\|) \frac{\vec{\mathbf{M}}_i}{\|\vec{\mathbf{M}}_i\| + \epsilon}$.

Particularly, $w_l^{10}, w_l^{01}, w_l^{11} \in \mathbb{R}$ are the learnable coefficients and $k_l^{10}, k_l^{01}, k_l^{11}$ are the steerable kernel bases that transform irreps from type 1 to 0, type 0 to 1, and type 1 to 1, respectively (c.f. Table 8 in [12] for more details); $\mathrm{MLP}_2(\mathbf{m}_{ij}) \in \mathbb{R}, \mathrm{MLP}_4(\|\vec{\mathbf{M}}_i\|) \in \mathbb{R}$. It is proved that the above implementation is equivariant with respect to the subgroup $SO_{\vec{\mathbf{g}}}(3)$.

We compare Steer-SE(2)-GNN with EGNN, EGNN-S (the subequivariant version of EGNN), and SGNN on Physion in Table 9.

Table 9: Comparison with Steerable SE(2) GNN.

|  | Dominoes | Contain | Link | Drape | Support | Drop | Collide | Roll |
|---|---|---|---|---|---|---|---|---|
| EGNN [10] | 61.3 | 66.0 | 52.7 | 54.7 | 60.0 | 63.3 | 76.7 | 79.8 |
| Steer-SE(2)-GNN [12] | 59.1 | 66.7 | 54.0 | 51.1 | 62.5 | 66.7 | 77.0 | 77.3 |
| EGNN-S | 72.0 | 64.6 | 55.3 | 55.3 | 60.5 | 69.3 | 79.3 | 81.6 |
| SGNN | 89.1 | 78.1 | 73.3 | 60.6 | 71.2 | 74.3 | 85.3 | 84.2 |

Steer-SE(2)-GNN outperforms EGNN on 5 out of 8 tasks and obtains comparable results on the other 3 tasks, which indicates the reliability of our implementation. The reason why Steer-SE(2)-GNN is generally better than EGNN lies in the involvement of the gravity constraint. If considering this constraint as well, EGNN-S consistently surpasses Steer-SE(2)-GNN. Overall, our SGNN achieves the significantly best performance.

# D  More Visualizations

Please refer to our Supplementary Video, presented at our project page `https://hanjq17.github.io/SGNN/`.

Moreover, to further evaluate the generalization of different models toward unseen scenes and assess whether they properly learn the effect of gravity, we conduct extra experiments by applying a rotation around a non-gravity axis, resulting in such scenarios where dominoes are placed on an incline while gravity still points downwards vertically.

The video is also presented at our project page. It is worth noticing that all models are only trained with the original data (horizontal table with vertical gravity) and none of them have seen any scenario placed like these. We have the following observations.

- Very interestingly, SGNN well generalizes to these novel scenarios and reasonably simulates the effect of gravity. Particularly, the domino at the bottom starts to slide down along the table driven by gravity. The dominoes at the top reach an equilibrium between friction and gravity and keep still. The small bottle placed on the table also falls down due to gravity.
- EGNN, as an E(3)-equivariant model, does not perceive the changes in scenarios, producing the same trajectory as if the table is horizontal. GNS and DPI, by not incorporating rotation symmetry, do not properly learn the effect of gravity as well.

This experiment interestingly reveals that our SGNN is able to learn how gravity acts on physical dynamics effectively from data and can thus generalize to novel scenes, verifying the validity of our motivation and design of subequivariance.

# E   More Insights on Subequivariance

In the paper we term subequivariance as a *relaxation* of equivariance. In order to help understanding the position of our work, we provide more explanations and comparisons between full-equivariant models, non-equivariant models, and our subequivariant models.

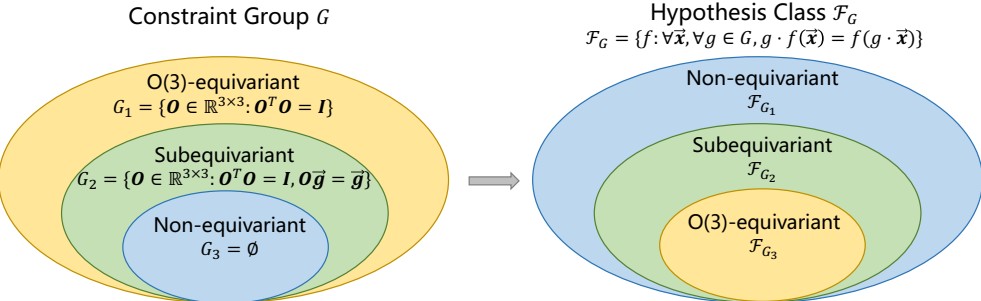

Figure 11: The comparison between O(3)-equivariant, subequivariant, and non-equivariant models.

As depicted in Fig. 11, equivariance is exerted by a group $G$ that serves as a constraint over the corresponding possible functions, or so called the hypothesis class $\mathcal{F}_G$. The functions in $\mathcal{F}_G$ must satisfy that for any group element $g \in G$, the output of the function should be transformed the same way as the input by $g$, or formally, $\mathcal{F}_G = \{f : \forall \vec{x}, g \in G, g \cdot f(\vec{x}) = f(g \cdot \vec{x})\}$. Particularly, if two groups satisfy $G_1 \subseteq G_2$, it is straightforward to see that the corresponding hypothesis class $\mathcal{F}_{G_2} \subseteq \mathcal{F}_{G_2}$.

Specifically, we consider O(3)-equivariant models with the constraint group $G_1 = \{O \in \mathbb{R}^{3 \times 3} : O^\top O = I\}$ including all orthogonal matrices, and non-equivariant models with $G_3 = \emptyset$. Clearly the non-equivariant models possess a larger hypothesis class, which are usually easier to optimize during training. However, the drawback is the weaker generalization since the optimized function might not obey the proper constraint that is implied in the data. This is experimentally verifies by the training and validation curve of DPI in Fig. 10. The O(3)-equivariant models, on the other hand, always satisfy the constraint $G$ and thus have a much smaller $\mathcal{F}$. In the existence of gravity, the symmetry is violated in the vertical direction, and not all $g \in$ O(3) should still serve as a constraint, but only those among $G_2 = \{O \in \mathbb{R}^{3 \times 3} : O^\top O = I, O\vec{g} = \vec{g}\}$. Therefore, $\mathcal{F}_{G_1}$ becomes over-constrained, which significantly impedes the training (see the training curve of EGNN in Fig. 10). Our subequivariant model, instead, leverages $G_2$ as the constraint with $G_3 \subseteq G_2 \subseteq G_1$ and therefore $\mathcal{F}_{G_1} \subseteq \mathcal{F}_{G_2} \subseteq \mathcal{F}_{G_3}$. With this proper relaxation, we expect the subequivariant models, equipped with the appropriate constraint, to have an ideal trade-off between training and generalization, which is also verified by our favorable experimental results of SGNN and the learning curve in Fig. 10.