# OpenReview forum: "Learning Physical Dynamics with Subequivariant Graph Neural Networks"
_NeurIPS.cc/2022/Conference — NeurIPS 2022 Accept_

### Official Review · Reviewer_bh6f · 2022-07-06

**Rating:** 6
**Confidence:** 4
**Soundness:** 3 good
**Presentation:** 2 fair
**Contribution:** 3 good

**Summary:**

The paper introduces a new model for learning physical dynamics. It introduces a concept that is termed subequivariance, which appears similar to restricted representations from group theory. The paper also introduces a hierarchical message passing to better enable long range interactions between particles. The paper also expands the input feature space by computing geometric features of the input particles and their objects. The paper experiments on a range of dynamics datasets, namely Physion and RigidFall.

**Questions:**

* How does subequivariance differs from using the restricted representation?
* How does including object information differ from being a specific feature space choice?
* How does the choice of using different edges to pass between objects/particles differ from an automorphism equivariant network?
* Why are only rotations applied around the gravity axis in the experiments?

**Strengths And Weaknesses:**

Strengths:
* The paper adds a "binary operation" which is the difference in position between adjacent features which adds a translation invariant representation to the input feature space, although the benefits and justification for this are not adequately detailed.
* The experiments seem thorough and multiple baselines are used, although I am not famailir with these baselines so I cannot comment on whether any relevant baselines have been missed.

Weaknesses:

[major]
* A key concept of the paper is characterising "subequivariance" as a relaxation of equivariance focusing on the group E(3). Although this concept has already been considered previously in [1,2] by using restricted representations of the group rather than the regular representation.
* A criticism of GNNs made in the paper is that they seldom explicitly involve the object information into the message passing, but this feels like a specific decision choice of the input features rather than some fundamental issue of GNNs which a paper can solve. Also, there are papers which consider specific object information such as [3].
* The concept of having different edges which pass different information appears strongly related to a relevant sub-field of GNNs exploring automorphism equivariance which has been ignored in this paper [4,5,6].
* In the experiments only rotations around the gravity axis are used. Surely, all rotations should be possible and all the models which do not correctly account for the break in symmetry caused by gravity should produce unrealistic results, while if this model works correctly it should break the symmetry and produce realistic results. I feel like this would be a far stronger result proving the correctness of the model.

[minor]
* Multiple typos / grammatical mistakes makes reading the paper difficult. (L44, 59)
* The object features appear to be the pooled features of its particles, which is used to create a feature space by subtracting these from each of the particles features. This seems to be a complexly worded way of saying subtracting the mean from the particle features which is a common approach in ML.


[1] Weiler, M. and Cesa, G., 2019. General e (2)-equivariant steerable cnns. Advances in Neural Information Processing Systems, 32.
[2] Cesa, G., Lang, L. and Weiler, M., 2021, September. A Program to Build E (N)-Equivariant Steerable CNNs. In International Conference on Learning Representations.
[3] Pfaff, T., Fortunato, M., Sanchez-Gonzalez, A. and Battaglia, P.W., 2020. Learning mesh-based simulation with graph networks. arXiv preprint arXiv:2010.03409.
[4] de Haan, P., Cohen, T.S. and Welling, M., 2020. Natural graph networks. Advances in Neural Information Processing Systems, 33, pp.3636-3646.
[5] Thiede, E., Zhou, W. and Kondor, R., 2021. Autobahn: Automorphism-based graph neural nets. Advances in Neural Information Processing Systems, 34, pp.29922-29934.
[6] Mitton, J. and Murray-Smith, R., 2021. Local Permutation Equivariance For Graph Neural Networks. arXiv preprint arXiv:2111.11840.

---

> ### Author Response · Authors · 2022-08-02
> **Response to Reviewer bh6f (Page 1/3)**
>
> We thank the reviewer for the valuable comments, and for pointing out several relevant references. We have cited them and discussed the differences.
> We still would like to emphasize that our work focuses on learning physical dynamics using GNNs, which is fundamentally different from the mentioned works. Specifically, [1,2] are steerable CNNs. [4,5,6] are GNNs but not designed for physical simulations. We kindly refer the reviewer to General Response for a description of the significant and unique challenges of physical simulation, especially on the complex physical prediction dataset like Physion.
> The concerns of the reviewer mainly stem from the insufficient discussion of the raised concepts in the mentioned works. We provide detailed responses to clarify the differences as follows.
>
>
> >**Q1. How does subequivariance differ from using the restricted representation?**
>
> We appreciate the reviewer for raising the two related papers [1,2] that have introduced the notion of group restriction. We have cited them and will discuss the differences in the following aspects:
>
> **1. Motivation.** While the papers [1,2] aim to develop equivariant models given any E(3)/E(2) and their subgroups, this paper is mainly interested in how the external force field relaxes the equivariance from an E(3)-equivariant model. These two scenarios can be correlated, if the subgroup induced by the external force field is tractable to derive. However, when the force field is complicated (distributed nonuniformly in space), the group restriction methods [1,2] might be no longer applicable, as the underlying subgroup is hard to derive. In contrast, based on the view of the force field, our method can still work.
>
> **2. Design.**  We follow the scalarization strategy used in EGNN to derive equivariant models and further augment the input with the external force vector to enable subequivariance. As shown in Eq. (9) of the main paper, our method is convenient to implement by typical computation of inner-product and nonlinearity. More importantly, we have proved the approximation universality in theory. For the methods like [1,2], they resort to steerable kernel bases in the form of harmonics and under irreducible group representations. Their components (including convolution and nonlinearity) should be specially designed. Another minor point is that the works [1,2] focus initially on CNN instead of GNN.
>
> **3. Performance.** To better illustrate, we also implement a baseline that leverages the idea in [1,2] but extends from CNN to GNN. Indeed, [1,2] are steerable CNNs, and these works have **not** offered available implementations on GNNs. We have tried our best to compare this idea with our model. Specifically, we implement ''Steerable-SE(2)-GNN'', which iterates the message passing as specified below. Consider the message computation for the edge $e_{ij}\in\mathcal{E}$ connecting node $i$ and $j$.
> * Compute the translation-invariant radial vector: $\vec{\mathbf{x}}_{ij}=\vec{\mathbf{x}}_i - \vec{\mathbf{x}}_j$.
> * Project $\vec{\mathbf{x}}\_{ij}$ onto $\vec{\mathbf{g}}$: $v\in\mathbb{R}=\frac{\vec{\mathbf{x}}\_{ij}\cdot\vec{\mathbf{g}}}{\|\vec{\mathbf{g}} \|}$, and $\vec{\mathbf{u}}\in\mathbb{R}^2=((\vec{\mathbf{x}}\_{ij}-v\vec{\mathbf{g}})\cdot \vec{\mathbf{m}}, ((\vec{\mathbf{x}}\_{ij}-v\vec{\mathbf{g}})\cdot \vec{\mathbf{n}})$, where $\vec{\mathbf{m}}, \vec{\mathbf{n}}$ are two orthonormal bases vertical to $\vec{\mathbf{g}}$.
> * Derive the type-0 message as $\mathbf{m}\_{ij} = \text{MLP}\_1(\sum\_l w^{01}\_l k^{01}\_l(\vec{\mathbf{u}}) \cdot \vec{\mathbf{u}}, v, \|\vec{\mathbf{u}}\|, \mathbf{h}\_i, \mathbf{h}\_j)$.
> * Derive the type-1 message as $\vec{\mathbf{M}}\_{ij}=(\sum\_l w\_l^{10} k^{10}\_l(\vec{\mathbf{u}})\mathbf{m}\_{ij}+\sum\_l w\_l^{11} k\_l^{11}(\vec{\mathbf{u}})\cdot\vec{\mathbf{u}})\cdot\text{MLP}\_2(\mathbf{m}\_{ij})$.
> * Aggregate and update type-0 feature: $\mathbf{h}'\_i=\text{MLP}\_3(\sum\_{j\in \mathcal{N}(i)}\mathbf{m}\_{ij}, \mathbf{h}\_i)$.
> * Aggregate and update type-1 feature: $\vec{\mathbf{M}\_i}=\sum\_{j\in\mathcal{N}(i)}\vec{\mathbf{M}}\_{ij}$,  $\vec{\mathbf{x}}\_{i}'=\vec{\mathbf{x}}\_{i} + \text{MLP}\_4(\|\vec{\mathbf{M}}\_{i}\|)\frac{\vec{\mathbf{M}}\_{i}}{\|\vec{\mathbf{M}}\_{i}\|+\epsilon}$.
>
> Particularly, $w\_l^{10}, w\_l^{01}, w\_l^{11}\in\mathbb{R}$ are the learnable coefficients and $k^{10}\_l, k^{01}\_l, k^{11}\_l$ are the steerable kernel bases that transform irreps from type 1 to 0, type 0 to 1, and type 1 to 1, respectively (please refer to Table 8 in [1] for more details); $\text{MLP}\_2(\mathbf{m}\_{ij})\in\mathbb{R}, \text{MLP}\_4(\|\vec{\mathbf{M}}\_{i}\|)\in\mathbb{R}$. It is proved that the above implementation is equivariant with respect to the subgroup $O\_{\vec{\mathbf{g}}}(3)$. We provide more explanations in Appendix.

---

> > ### Author Response · Authors · 2022-08-02
> > **Response to Reviewer bh6f (Page 2/3)**
> >
> > We compare Steer-SE(2)-GNN with EGNN, EGNN-S (the subequivariant version of EGNN), and SGNN on Physion:
> >
> > |                | Dominoes    | Contain     | Link        | Drape       | Support     | Drop        | Collide     | Roll        |
> > | -------------- | ----------- | ----------- | ----------- | ----------- | ----------- | ----------- | ----------- | ----------- |
> > | EGNN           | 61.3 | 66.0        | 52.7        | 54.7 | 60.0        | 63.3        | 76.7        | 79.8 |
> > | Steer-SE(2)-GNN | 59.1        | 66.7 | 54.0 | 51.1        | 62.5 | 66.7 | 77.0 | 77.3        |
> > | EGNN-S | 72.0     | 64.6     | 55.3     | 55.3     | 60.5     | 69.3     | 79.3     | 81.6     |
> > | SGNN           | **89.1**    | **78.1**    | **73.3**    | **60.6**    | **71.2**    | **74.3**    | **85.3**    | **84.2**    |
> >
> > Steer-SE(2)-GNN outperforms EGNN on 5 out of 8 tasks and obtains comparable results on the other 3 tasks, which indicates the reliability of our implementation. The reason why Steer-SE(2)-GNN is generally better than EGNN lies in the involvement of the gravity constraint. If considering this constraint as well, EGNN-S surpasses Steer-SE(2)-GNN on 6 tasks. Overall, SGNN achieves the significantly best performance.
> >
> > We have added citations of the mentioned works [1,2] in the paper.
> >
> > [1] Weiler, M. and Cesa, G., 2019. General e (2)-equivariant steerable cnns.
> >
> > [2] Cesa, G., Lang, L. and Weiler, M., 2021, September. A Program to Build E (N)-Equivariant Steerable CNNs.
> >
> > > **How does including object information differ from being a specific feature space choice?**
> >
> > There is probably some misunderstanding here. Our design of involving object information in the message passing has fundamental differences from a simple feature choice.
> > **(1)** We incorporate object information **in two aspects**: the geometric information ($\vec{\mathbf{C}}$) and the scalar information ($\mathbf{c}$). To ensure equivariance, these two pieces of information should be treated **differently** as depicted in our message passing (Eq. (5-6)). By contrast, in the implementation of GNS/DPI by Physion and the raised work [3] by the reviewer, this information is simply concatenated together to the node feature, in a way closer to being a "specific feature space choice".
> > **(2)** The object information, in our design, is constantly updated during the hierarchical modeling. Specifically, $\vec{\mathbf{C}}$ and $\mathbf{c}$ are updated in inter-object message passing (Eq. (11)). The **updated** $\vec{\mathbf{C}}'$ and $\mathbf{c}'$ are then used in the inner-object message passing.
> > **(3)** Our approach of adding the object information is theoretically guaranteed to enhance the expressivity of our message passing (SOMP) over EGNN and GMN. More specifically, in Appendix A.3 we theoretically reveal that both EGNN and GMN are special cases of SGNN by choosing specific forms of MLP in $\phi_{\vec{\mathbf{g}}}$ and $\psi_{\vec{\mathbf{g}}}$ in Eq (7-8). This also supports that our way of involving object information is not an arbitrary feature space choice.
> >
> > [3] Pfaff et al, 2020. Learning mesh-based simulation with graph networks. arXiv preprint arXiv:2010.03409.

---

> > > ### Author Response · Authors · 2022-08-05
> > > **Response to Reviewer bh6f (Page 3/3)**
> > >
> > > > **How does the choice of using different edges to pass between objects/particles differ from an automorphism equivariant network?**
> > >
> > > In the raised papers [4,5,6], an automorphism is defined as an isomorphism from a graph to itself (see Eq. (1) in [4]). This concept seems unrelated to edge separation. Natural Graph Networks (NGN) [4] do allow different message passing kernels on non-isomorphic edges, which, however, is clearly different from our motivation. Inspired by physics, we use different edges to simulate different interactions between or within objects, without the consideration of which edges are isomorphic and which are not. The models developed by [4,5,6] do not explicitly discuss different message passing between objects and particles.
> > >
> > > [4] de Haan, P., Cohen, T.S. and Welling, M., 2020. Natural graph networks.
> > >
> > > [5] Thiede, E., Zhou, W. and Kondor, R., 2021. Autobahn: Automorphism-based graph neural nets.
> > >
> > > [6] Mitton, J. and Murray-Smith, R., 2021. Local Permutation Equivariance For Graph Neural Networks.
> > >
> > > > **Why are only rotations applied around the gravity axis in the experiments?**
> > >
> > > As we are focusing on predicting the dynamics of physical scenes where gravity exists, it is natural to leverage the gravity direction as inductive bias. As shown in Physion and RigidFall, the rotated scene can be easily generated from the original samples by restricting the rotation around gravity. However, if we try an arbitrary rotation (not just around gravity), the gravity direction should also rotate correspondingly. Otherwise, for example, the gravity could be parallel to the ground, leading to novel scenes that are unseen in the dataset. Without a ground-truth scene, it is hard to justify if the models produce realistic results and behave correctly. Still, it is worth mentioning that our formulation of SGNN is capable of taking the arbitrary direction of the external force field to meet subequivariance.

---

> > > > ### Comment · Reviewer_bh6f · 2022-08-07
> > > > **Response**
> > > >
> > > > I think the theoretical contribution of natural graph networks is very similar to yours. Having a different message passing kernel for non-isomorphic edges is the same as having a different edge between different objects. The isomorphism group of the edge is driven by the neighbouring objects. I don't see the difference from the description provided, please elaborate. If they are the same, then this could just be recognised, and the novelty of the paper comes in creating a specific network for the tasks considered.
> > > >
> > > > I think I already said I am not familiar with the given tasks. Despite this I can only assume dominoes is say a set of dominoes set up and then pushed over or something like this and the task is to predict the dynamics of this. Rotations around the gravity axis leave the task identical, just in a different rotation. Surely you could still have a small rotation around a non-gravity axis, which would amount to having the dominoes placed along a slight incline, which gravity still acts vertically. This was what I was trying to suggest, would this be possible? I would be interested to see how each model generalized to these settings.

---

> > > > > ### Author Response · Authors · 2022-08-08
> > > > > **Further Responses (Part 3)**
> > > > >
> > > > > > **I think the theoretical contribution of natural graph networks is very similar to yours....If they are the same, then this could just be recognised, and the novelty of the paper comes in creating a specific network for the tasks considered.**
> > > > >
> > > > > We agree that the edge separation has some correlations to natural graph networks, in that they both aim to incorporate different kernels for different types of interactions. In this sense, SGNN is a specific and effective instantiation for the physical simulation task, with the help of our designed hierarchical message passing scheme. We will add the above explanations to the revised version.
> > > > >
> > > > >
> > > > >
> > > > > > **Surely you could still have a small rotation around a non-gravity axis, which would amount to having the dominoes placed along a slight incline, which gravity still acts vertically. This was what I was trying to suggest, would this be possible? I would be interested to see how each model generalized to these settings.**
> > > > >
> > > > > Sorry for the misunderstanding. Thank you very much for further elaborating on this point. We agree that this is a very meaningful experiment for evaluating whether these models actually learn the effect of gravity. In light of this, we conduct extra experiments by applying a rotation around a non-gravity axis, resulting in such scenarios where dominoes are placed on an incline while gravity still points downwards vertically.
> > > > >
> > > > > To facilitate viewing, the demonstrations are provided in `rotate.mp4` in this [anonymous link](https://drive.google.com/drive/folders/1NRBfwNk9yLMNii88ep0c0U8GquMKQeUb?usp=sharing). It is also worth noticing that all models are only trained with the original data (horizontal table with vertical gravity) and none of them have seen any scenario placed like these. We have the following observations.
> > > > >
> > > > > * Very interestingly, SGNN well generalizes to these novel scenarios and reasonably simulates the effect of gravity. Particularly, the domino at the bottom starts to slide down along the table driven by gravity. The dominoes at the top reach an equilibrium between friction and gravity and keep still. The small bottle placed on the table also falls down due to gravity.
> > > > > * EGNN, as an E(3)-equivariant model, does not perceive the changes in scenarios, producing the same trajectory as if the table is horizontal. GNS and DPI, by not incorporating rotation symmetry, do not properly learn the effect of gravity as well.
> > > > >
> > > > > This experiment interestingly reveals that our SGNN is able to learn how gravity acts on physical dynamics effectively from data and can thus generalize to novel scenes, verifying the validity of our motivation and design of subequivariance. We are willing to provide the above demo and explanations in the revised paper.
> > > > >
> > > > >
> > > > > *Thanks again for your insightful suggestions. Please do not hesitate to contact us if there are other clarifications or experiments we can offer.*
> > > > >
> > > > > *Thank you for your time!*
> > > > >
> > > > >
> > > > > Best, \
> > > > > Authors

---

> > > > > > ### Comment · Reviewer_bh6f · 2022-08-09
> > > > > > **Response**
> > > > > >
> > > > > > I personally think this is quite an impactful result, although it is expected that the other models would perform poorly. It would have been good to see the extra model you added (steer-SE(2)-GNN), given that if implemented correctly it should handle gravity well. I can acknowledge this is maybe a pain though and you have already demonstrated your method works here.
> > > > > >
> > > > > > I think my only main issue remaining is how much extra conversation was required to understand the approach and how much of a re-write is required so that the paper is easy to understand. In addition, the extra writing to compare to all the different approaches within this conversation as I think this is really important to understand the novelty and contribution of this paper.

---

> > > > > > > ### Author Response · Authors · 2022-08-09
> > > > > > > **Thank you for recognizing our contributions!**
> > > > > > >
> > > > > > >
> > > > > > > > **I personally think this is quite an impactful result, although it is expected that the other models would perform poorly.**
> > > > > > >
> > > > > > > Thank you for the supportive comments! We agree that this result strengthens the validity of the model, especially on the design of subequivariance. We have updated this observation in the revised manuscript. Please refer to our response below for a detailed list of revisions we have made to enhance the readability of the paper.
> > > > > > >
> > > > > > > > **I think my only main issue remaining is how much extra conversation was required to understand the approach and how much of a re-write is required so that the paper is easy to understand. In addition, the extra writing to compare to all the different approaches within this conversation as I think this is really important to understand the novelty and contribution of this paper.**
> > > > > > >
> > > > > > > Thank you for the comment! We agree that there are several important points mentioned in these discussions with you that are vital for the readers to better understand the contributions of this paper. To reflect these crucial contents in this discussion, we have made the following revisions to the paper:
> > > > > > >
> > > > > > > * We add discussions to those leveraging restricted representations to obtain equivariance on subgroups [1, 2] as well as general approaches like EMLP [3] in a separate paragraph in Related Work. We also discuss the differences between our approach and these works. (Line 44-48)
> > > > > > > * We update the results and discussions of our implemented Steer-SE(2)-GNN in Experiment. (Line 310-313)
> > > > > > > * We clarify the careful considerations on incorporating object information into the message passing. This part better clarifies how our approach differs from a simple feature space choice. (Line 175-179)
> > > > > > > * We add the connection between edge separation and automorphism graph networks [4, 5, 6]. (Line 200-203)
> > > > > > > * We update the experiment of generalization toward rotations along a non-gravity axis in Experiment. (Line 324-333)
> > > > > > >
> > > > > > > There are also corresponding revisions in Appendix that provide necessary details of these updated contents, e.g., the implementation of our adapted Steer-SE(2)-GNN.
> > > > > > >
> > > > > > > We sincerely thank the reviewer for recognizing the novelty of this paper and having these discussions with fruitful suggestions! We hope the revisions we made to the manuscript according to these discussions have improved the presentation and made it easier for readers to understand. Please do not hesitate to contact us if there are other clarifications we can offer.
> > > > > > >
> > > > > > > Thank you for your time!
> > > > > > >
> > > > > > > Best, \
> > > > > > > Authors
> > > > > > >
> > > > > > > [1] Weiler, M. and Cesa, G., 2019. General e (2)-equivariant steerable cnns. Advances in Neural Information Processing Systems, 32.
> > > > > > >
> > > > > > > [2] Cesa, G., Lang, L. and Weiler, M., 2021, September. A Program to Build E (N)-Equivariant Steerable CNNs. In International Conference on Learning Representations.
> > > > > > >
> > > > > > > [3] Finzi et al. A Practical Method for Constructing Equivariant Multilayer Perceptrons for Arbitrary Matrix Groups. In ICML.
> > > > > > >
> > > > > > > [4] de Haan, P., Cohen, T.S. and Welling, M., 2020. Natural graph networks. Advances in Neural Information Processing Systems, 33, pp.3636-3646.
> > > > > > >
> > > > > > > [5] Thiede, E., Zhou, W. and Kondor, R., 2021. Autobahn: Automorphism-based graph neural nets. Advances in Neural Information Processing Systems, 34, pp.29922-29934.
> > > > > > >
> > > > > > > [6] Mitton, J. and Murray-Smith, R., 2021. Local Permutation Equivariance For Graph Neural Networks. arXiv preprint arXiv:2111.11840.

---

> > > > > > > > ### Comment · Reviewer_bh6f · 2022-08-09
> > > > > > > > **Response**
> > > > > > > >
> > > > > > > > Thank you for the discussion. I will revise my score.

---

> > > > > > > > > ### Author Response · Authors · 2022-08-09
> > > > > > > > > **Thank you for the support!**
> > > > > > > > >
> > > > > > > > > Dear Reviewer bh6f,
> > > > > > > > >
> > > > > > > > > Thank you very much! We really enjoy the discussion with you, during which your insightful comments have helped greatly improve the paper. Thanks again!
> > > > > > > > >
> > > > > > > > > Best, \
> > > > > > > > > Authors

---

> > > ### Comment · Reviewer_bh6f · 2022-08-07
> > > **Response**
> > >
> > > I agree that what you are doing is not just an arbitrary feature space choice and if I suggested that I apologize as that is unfair.
> > >
> > > On the other hand, it seems like you have vector feature spaces and scalar feature spaces. Therefore, they clearly should be treated differently and not just concatenated together. Am I correct in thinking that just concatenating them would be the same as symmetry breaking, because you would just be ignoring the symmetry associated with the vector features? How does this differ from say a rotation equivariant network, where the first irrep is the scalar feature and the other irreps are the vector features? I am trying to assess how novel the splitting of geometric information and scalar information is as it seems strongly related to other equivariant approaches.
> > >
> > > I do agree there is some novelty in creating a new message passing network for a specific application though.

---

> > > > ### Author Response · Authors · 2022-08-08
> > > > **Further Responses (Part 2)**
> > > >
> > > > > **Am I correct in thinking that just concatenating them would be the same as symmetry breaking, because you would just be ignoring the symmetry associated with the vector features?**
> > > >
> > > > Yes. If one simply concatenates them (like as done by GNS and DPI), equivariance will be no longer maintained, and this is indeed why scalar feature and vector features should be treated differently.
> > > >
> > > > > **How does this differ from say a rotation equivariant network, where the first irrep is the scalar feature and the other irreps are the vector features?**
> > > >
> > > > The irreps-based rotation equivariant networks resort to computing spherical harmonics which is very computationally expensive. Moreover, the hidden dimension grows significantly with the order of irreps feature computed in the network.
> > > >
> > > > Our method of ensuring equivariance belongs to the scalarization family which leverages the inner-product of geometric vectors [1, 2]. The method is easy to implement, highly efficient, and has better scalability toward large systems than the irreps-based equivariant networks. Interestingly, our approach also comes with a strong theoretical guarantee of universality, as depicted in Theorem 1 and Appendix A.1.
> > > >
> > > > [1] Satorras et al. E(n)-equivariant graph neural networks. ICML 2021.
> > > >
> > > > [2] Villar et al. Scalars are universal: Equivariant machine learning, structured like classical physics. NeurIPS 2021.

---

> > ### Comment · Reviewer_bh6f · 2022-08-07
> > **Response**
> >
> > Thanks for your response. I am not trying to state that the task you are solving is not challenging, but solely focusing around the technical details of the proposed model to identify the novelty.
> >
> > 1. Is the external force field not tractable to calculate? I assume in the case of gravity this is straight forward as it is just breaking the rotational symmetry that is not around the z-axis. Also, if there is a non-uniformly distribution forcefield in the space that breaks rotational symmetry I don't follow what symmetry would be left, could you please elaborate and provide an intuitive example of this?
> >
> > 2. I agree a more general method for finding the symmetries is beneficial. How does your method compare to the EMLP approach, which is also a general approach to building equivariant networks:
> > https://emlp.readthedocs.io/en/latest/#
> >
> > 3. Thank you for considering the additional experiments. It is quite surprising that the new model performs very similarly to the EGNN approach despite correctly modelling the symmetry. What is the key difference between your model and this new one which leads to such a drastic difference in performance?

---

> > > ### Author Response · Authors · 2022-08-08
> > > **Further Responses (Part 1)**
> > >
> > > Dear Reviewer bh6f,
> > >
> > > We really enjoy communicating with you and appreciate your efforts. We provide detailed explanations to your raised questions below.
> > >
> > > > **Is the external force field not tractable to calculate?  I assume in the case of gravity this is straightforward as it is just breaking the rotational symmetry that is not around the z-axis.**
> > >
> > > Yes, the symmetry implied in the external force field might be difficult to infer especially when it is not uniformly distributed. Moreover, some fields only possess local symmetry other than a global symmetry. We will give detailed examples in the answer to the next question.
> > >
> > > > **Also, if there is a non-uniformly distribution force field in the space that breaks rotational symmetry I don't follow what symmetry would be left, could you please elaborate and provide an intuitive example of this?**
> > >
> > > We are glad to offer more explanations. Indeed, our theory has more implications and is not restricted to uniformly distributed force fields like gravity. Particularly, the force field can possess local symmetry even when there is no clear global symmetry. Here is an example.
> > >
> > > Consider a ball rolling on a smooth and non-flat slope (which can be modeled as a manifold), under the effect of gravity. Due to the non-uniform curvature on the manifold, the tangent
> > > force field (formulated by the resultant force of gravity and the support force) is also non-uniform. Clearly, such field is not globally symmetric.
> > >
> > > Now, consider there is a locally flat surface on the manifold. It is straightforward to see that the movement of the ball within this region follows symmetry (translation and rotation). This example illustrates that certain local symmetries possibly exist in force fields, and should also be captured by the model.
> > >
> > > It is hard (at least exhausting) to derive the symmetry for each point in the manifold. Hence the group restriction methods are NOT easy to apply. Instead, our formulation well covers this scenario, by making use of the force field $\vec{\mathbf{g}}$ as an extra input, where $\vec{\mathbf{g}}$ does not necessarily need to be globally equivariant.
> > >
> > > > **I agree a more general method for finding the symmetries is beneficial. How does your method compare to the EMLP approach.**
> > >
> > > Thanks for raising this point. Our approach differs from EMLP in the following aspects:
> > >
> > > * EMLP constructs equivariant MLPs for arbitrary matrix groups by solving the group constraint. Our approach tackles E($n$) group and its subgroups like $O_{\vec{\mathbf{g}}}(3)$ by leveraging the force field vector ${\vec{\mathbf{g}}}$.
> > > * Our approach is easier to implement and is computationally more efficient than EMLP. Specifically, EMLP requires solving the group constraints using SVD, yielding a time complexity of $O(n^3)$ where $n$ is the input dimension. By contrast, our approach mainly requires to compute the inner product which scales linearly with the input dimension $n$.
> > > * EMLP requires specifying the detailed group to perform equivariance constraints. Our approach only needs to take as input the force field vector $\vec{\mathbf{g}}$. For those scenarios when the group is intractable to specify (like the example of local symmetry we provided in the previous question), our approach is more flexible.
> > >
> > > > **What is the key difference between your model and this new one which leads to such a drastic difference in performance?**
> > >
> > > Our model features more proposed elements that are verified to be also very important in our ablation study, including considering the object geometric information as well as the hierarchical message passing scheme. Unfortunately, it is non-straightforward to incorporate all these components in the new model "Steer-SE(2)-GNN", which could explain the drastic difference in performance.

---

> > > > ### Comment · Reviewer_bh6f · 2022-08-09
> > > > **Response**
> > > >
> > > > Thank you for the clarifications. I think I am slowly better understanding the approach.
> > > >
> > > > One further question, if time permits, how does you approach then generalise to a new input domain. Here I am thinking of the example you gave with the ball rolling on a smooth manifold, which has some local symmetries but no global symmetries. If the model trained on some of these examples is then tested on a new manifold, does it generalise well? Also, has this been shown in the paper (please point me to it if it has)? When generalising to a new domain such as this does the method require the input domain to be provided to the model so that it has the force field? How much cost is there in computing this force field? Finally, if you have this force field how would this approach compare to a more classical approach of solving the task where I just compute the forces on the object at each step and predict its motion (given I assume this would be possible due to having access to the force field)?
> > > >
> > > > I think providing more intuitive examples such as these discussed into the paper may help a reader understand the method (at least it has for me).

---

> > > > > ### Author Response · Authors · 2022-08-09
> > > > > **Further Responses**
> > > > >
> > > > >
> > > > > > **One further question, if time permits, how does you approach then generalise to a new input domain...If the model trained on some of these examples is then tested on a new manifold, does it generalise well? Also, has this been shown in the paper (please point me to it if it has)?**
> > > > >
> > > > > In the paper, experiments are conducted on physical scene data with gravity. We may not be able to evaluate such scenario where the force fields are highly irregular and non-uniform since the discussion period is running out of time. However, we do believe that our approach can generalize to these scenarios as it is designed to incorporate force field into the symmetry modeling.
> > > > >
> > > > > Besides, our experiment presented in the previous discussion (about rotation around a non-gravity axis) can actually be treated as one specific instantiation of this cross-domain generalization. In fact, as for our model, the setup in this experiment, which is rotating the scene while still keeping gravity vertical, is equivalent to keeping the scene as it is while rotating the direction of gravity by the same degree. In this sense, the previous results have implied that our model can generalize to a new domain (a new configuration of force field) different from the training data. We will keep on investigating more complicated scenarios that may go beyond simulations on physical scene tasks.
> > > > >
> > > > > > **When generalising to a new domain such as this does the method require the input domain to be provided to the model so that it has the force field? How much cost is there in computing this force field?**
> > > > >
> > > > > Yes, we require the force field to be provided. Learning the force field itself is indeed another challenging task. Our approach here is to model the effect of a given force field on the physical dynamics of multiple interacting objects.
> > > > >
> > > > > > **Finally, if you have this force field how would this approach compare to a more classical approach of solving the task where I just compute the forces on the object at each step and predict its motion (given I assume this would be possible due to having access to the force field)?**
> > > > >
> > > > > The biggest challenge here is that even if the force field is given, how it acts on the dynamics of objects still remains unknown and very challenging to learn. Particularly, apart from the force field, there are also internal forces of physical systems, like friction, collision, support, etc. The dynamics of an object are influenced by not only the force field but these internal forces as well. Our model is designed to learn the complicated effect of **both force field and internal forces** between objects to accurately predict the dynamics.
> > > > >
> > > > > Here is an example. Imagine there is a cube placed on a table. The cube is affected by both gravity and the support, and it remains still. However, if the table is removed, the cube will fall since the support force no longer exists. This implies that simple approaches like `just compute the forces on the object at each step and predict its motion` cannot generalize, since the object is affected by a combination of force field and interactions with other objects, and the interactions with other objects are not provided, which should be learned by the model.

---

> ### Author Response · Authors · 2022-08-06
> **Looking forward to your post-rebuttal feedback**
>
> Thanks again for your insightful suggestions and comments. As the deadline for discussion is approaching, we are glad to provide any additional clarifications that you may need.
>
> In our previous response, we have carefully studied your comments and added a lot more experiments and analyses to complement your suggestions. We summarize our responses with regard to the following aspects:
>
> * We clearly discuss the fundamental differences between this paper and those using restricted representations, from motivation, design, to performance. We have also, to our best effort, adapted and included an additional baseline to compare with this line of work.
> * We elaborate on the differences between our way of injecting object geometric information and that of a pure feature space choice.
> * We explain the physics-driven motivation and implementation of our edge separation, which clearly distinguishes it from graph automorphism.
> * Reasons for rotating along the gravity axis: for better leveraging the data for the models and baselines while ensuring physical correctness of symmetry.
>
> We hope that the provided new experiments and additional explanations have convinced you of the merits of our work. Please do not hesitate to contact us if there are other clarifications or experiments we can offer.
>
> Thank you for your time again!
>
> Best,
>
> Authors

---

### Official Review · Reviewer_cLqG · 2022-07-12

**Rating:** 5
**Confidence:** 5
**Soundness:** 3 good
**Presentation:** 3 good
**Contribution:** 2 fair

**Summary:**

In this work, authors present a new formulation of the equivariant graph neural network, namely, subequivariant GNN (SGNN), which allows the modeling of systems with symmetry breakage such as gravity. In addition, to model systems with different shapes and geometry, an additional feature that represent the object type is added so as to distinguish the intra-object (elasticity/rigidity/plasticity) and inter-object interactions (collision/repulsion/weak attraction). A hierarchical modeling approach is implemented to address particle- and object-level interactions separately. By considering different physical scenarios, for instance, collision and contact prediction, the superior performance of SGNN is demonstrated.

**Questions:**

Some of the questions that naturally follow from the previous sections and some additional questions the authors should address are mentioned below.
1) Since one of the main aim of the present work is to simulate realistic contact models, additional baselines which employ contact models in the Lagrangian/Hamiltonian neural network framework may be considered. Indeed, they may not scale as the GNNs, but they can potentially give improved conservation of physical laws than purely data-driven approaches. Also, this will provide insights into the deficiencies of SGNN.
2) Authors have not evaluated how realistic the trajectory is with respect to the physical laws. For instance, is the energy and momentum conserved in the collision, is the coefficient of restitution 1, etc. These are important to analyze, since the claim is that the SGNN provides improved dynamics (Q2 in results).
3) Again, since the contact is of main concern in the present work, authors should evaluate how the model performs in situations where there is self contact. At present, it is not clear whether the formulation is capable of dealing with such scenarios.

**Limitations:**

There are several limitations for the present work, which the authors should consider.
1) Although a particle-based approach is employed, unlike previous works such GNS, there are no discussions on deformable systems. It is not clear how much improvement SGNN can give for deformable systems.
2) As mentioned in the questions, discussions on self-contact and how this is incorporated is missing.
3) Performance of SGNN on systems with drag and other dissipative forces are lacking.
4) In the case of a deformable system, consider a scenario where a particle from a given object breaks away, gets reflected by the wall, and then comes back to interact with the same object. In such case, if the particle comes with the \varepsilon cutoff distance, does SGNN model this as a contact or as a particle of the object. In other words, does the particle "heal" with the object or not. If yes, this is unphysical. It is not clear if the model can address this issue.

**Strengths And Weaknesses:**

Overall, the work is well-written and clearly presented. It builds on the equivariant GNNs and modifies it to present the idea of SGNN, which can include directional symmetry breaking such as gravity. TSome of the main comments regarding the work are as follows.
1) While the idea is useful, the implementation and proofs are fairly straightforward extension from the equivariant case. Also, no additional inductive biases to preserve the physics (as in the case of Hamiltonian or other physics-informed GNNs) are implemented. This raises a question on the validity of the trajectory predicted. Specifically, no comments on whether the trajectories represent a physically feasible realization is not discussed. This is important because one of the major advantages the authors claim for SGNN is the ability to "learn" the dynamics.
2) Authors refer to previous works such as Hamiltonian GNNs, and mention that they do not consider rotational equivariance. This is incorrect. In Hamiltonian GNNs, the edge embeddings can be modified to have the distance (L2 norm of the difference in positions), instead of giving simply the vectorial difference. Since the functional form to be learned in the case of a Hamiltonian is scalar, this approach also works very well and is both translationally and rotationally invariant.
3) The examples demonstrated in the work are those, where contact seems to be of main interest. It is not clear how much better the model would perform in other cases where contact is not necessarily the primary interest but dynamics is. Now, if the main focus of the work is simulate contact, then there has been several works which attempted to do this (for instance, Zhong, Y.D., Dey, B. and Chakraborty, A., 2021. Extending lagrangian and hamiltonian neural networks with differentiable contact models. Advances in Neural Information Processing Systems, 34, pp.21910-21922.), which employ a similar to idea of cutoff-based contact detection. Indeed, they do not employ a graph-based approach. However, in the present case although the particles are considered, there do not seem to be any deformation simulated and hence, these approaches referred should also stand equally valid as SGNN.
4) Also, the experiments chosen seem to be favorable for SGNN in comparison to the other baselines. For instance, implementing EGNN and GMN, exactly as they are, with gravity is expected to yield poor performance as the architecture expects the data to be rotationally invariant. Similarly, GNS and DPI learns purely from data and hence are unaware of symmetry unless trained specifically for it. More interesting examples where the situations where GNS and EGNN have been shown to yield SOTA performance could give a more realistic representation of how better SGNN is in comparison to these models.

---

> ### Author Response · Authors · 2022-08-02
> **Response to Reviewer cLqG (Page 1/3)**
>
> We thank the reviewer for the thoughtful comments. To make our response more compact, we have rearranged the questions and addressed the similar ones together.
>
>
>
> > **Q1. The originality: while the idea is useful, the implementation and proofs are fairly straightforward extension from the equivariant case.**
>
> We thank the reviewer for recognizing the usefulness of the idea. Our theoretical derivations of subequivariance does stem from the equivariant case. However, in Theorem 1, we have proved that our formuation of the subequivariant message passing has theorectical guarantee of universality, which broadly enhances the equivariant models when external force exists. The proof of Theorem 1 is not trivial compared to the traditional equivariance case (Proposition 1) as shown in Appendix A. For example, we require to additionaly prove the claim in Eq. (17) that states the one-to-one maping between the equivalent class $\{\mathbf{O}\vec{\mathbf{Z}}\mid \mathbf{O}\in O_{\vec{\mathbf{g}}}(3)\}$ and the augmented inner-product $[\vec{\mathbf{Z}}, \vec{\mathbf{g}}]^\top[\vec{\mathbf{Z}}, \vec{\mathbf{g}}]$.
>
>
>
> > **Q2. Discussion of physics-informed NNs.**
>
> > **Also, no additional inductive biases to preserve the physics (as in the case of Hamiltonian or other physics-informed GNNs) are implemented. This raises a question on the validity of the trajectory predicted. Specifically, no comments on whether the trajectories represent a physically feasible realization is not discussed. This is important because one of the major advantages the authors claim for SGNN is the ability to "learn" the dynamics.**
>
> > **Now, if the main focus of the work is simulate contact, then there has been several works which attempted to do this (for instance, Zhong, Y.D., Dey, B. and Chakraborty, A., 2021. Extending lagrangian and hamiltonian neural networks with differentiable contact models. Advances in Neural Information Processing Systems, 34, pp.21910-21922.).**
>
>
>
> We would like to first clarify the difference between SGNN and the other two lines of existing works: GNS/DPI, and the Hamiltonian-based NNs or other Physics-Informed NNs [1, 2].
>
>
> GNS and DPI are particle-based GNN simulators, belonging to the intuitive physics family. They usually do not explicitly involve hand-crafted conservations or equations (e.g., the symplectic update in Hamiltonian NNs), purely learning the dynamics from data, similar to how humans percieve, learn, and induce about the dynamics.
>
> Hamiltonian-based NNs and also other PINNs are differentiable physics models that explicitly build up physical equations into the model, restricting the output trajectory to possess some desirable physical properties, such as energy conservation.
>
> Our model SGNN belongs to the first category, but is designed to address some limitations of these models. Particularly, we inject mild physical prior like proper symmetry into the model, making it strongly generalizable and data-efficient (see Table 1 and 3 in the manucript).
>
> SGNN adds the inductive bias of subequivariance to reflect the partial symmetry of the physical law. In contrast to PINNs, we avoid over-restricted inductive bias to enable more generalization ability across different types of systems. For example, the Hamiltonian-based NNs are generally designed to pursue energy conservation, which, however, is usually broken by forces like friction between objects, for the case like Physion. In this sense, the way we add inductive bias into SGNN is well appropriate.
>
> We cite these mentioned works in the paper.
>
> [1] Sanchez-Gonzalez et al. Hamiltonian graph networks with ode integrators.
> [2] Zhong et al. Extending lagrangian and hamiltonian neural networks with differentiable contact models.
>
> > **Authors have not evaluated how realistic the trajectory is with respect to the physical laws.**
>
>
> To evaluate the validity of the predicted trajectory, we followed the settings in Physion and RigidFall, and applied quantities including contact accuracy and rollout MSE. The video demonstrations were also provided in the supplementary material, where SGNN did produce visually realistic trajectories.

---

> > ### Author Response · Authors · 2022-08-02
> > **Response to Reviewer cLqG (Page 2/3)**
> >
> > > **Q3. Experimental comparison with PINNs.**
> >
> > >**Additional baselines which employ contact models in the Lagrangian/Hamiltonian neural network framework may be considered. Indeed, they may not scale as the GNNs, but they can potentially give improved conservation of physical laws.**
> >
> > Thank you for this suggestion. We augment SGNN and EGNN by a Hamiltonian integrator. Details of the implementation include:
> >
> > **(1)** We leverage a sum-pooling over the output scalar feature ($\mathbf{h}\_i$) as the Hamiltonian of the system, i.e., $\mathcal{H} \in \mathbb{R}=\sum\_{i=1}^N \mathbf{h}\_i$.
> >
> > **(2)** We employ a RK1 integrator to conduct Hamiltonian update, i.e., $(\dot{\vec{\mathbf{q}}}, \dot{\vec{\mathbf{p}}})=(\frac{\partial\mathcal{H}}{\partial{\vec{\mathbf{p}}}}, -\frac{\partial\mathcal{H}}{\partial{\vec{\mathbf{q}}}})$.
> >
> > One thing worth noticing here is that we are assuming the particles possess uniform mass, so that $\vec{\mathbf{q}}, \vec{\mathbf{p}}$ can be derived from $\vec{\mathbf{x}}, \vec{\mathbf{v}}$, respectively. We name the variants as SGNN-H and EGNN-H ("H" stands for Hamiltonian), and evaluate them on Physion. The results are displayed in the following table.
> >
> >
> > |        | Domino | Contain  | Link     | Drape    | Support  | Drop     | Collide  | Roll     |
> > | ------ | -------- | -------- | -------- | -------- | -------- | -------- | -------- | -------- |
> > | EGNN   | 61.3     | 66.0     | 52.7     | 54.7     | 60.0     | 63.3     | 76.7     | 79.8     |
> > | EGNN-H | 52.0     | 58.1     | 54.0     | 54.3     | 51.1     | 54.7     | 75.7     | 75.3     |
> > | SGNN   | **89.1** | **78.1** | **73.3** | **60.6** | **71.2** | **74.3** | **85.3** | **84.2** |
> > | SGNN-H | 69.9     | 66.0     | 61.1     | 60.3     | 55.3     | 62.0     | 79.3     | 78.7     |
> >
> > Adding Hamiltonian into EGNN and SGNN generally leads to detrimental performance. We speculate that it is probably due to the dissipative forces as well as highly complex interactions in Physion. This result suggests that it may not be beneficial to involve such strong physical inductive bias for the scenarios in Physion, which also verify our discussion in Q2 above. By the way, SGNN-H is always better than EGNN-H.
> >
> > As the reviewer said, the Hamiltonian NNs may not scale as GNNs. The Hamiltonian module brings significant computation overhead during training. We list the average training time per step (in seconds) on Physion Dominoes dataset.
> >
> > | EGNN          | EGNN-H       | SGNN          | SGNN-H        |
> > | ------------- | ------------ | ------------- | ------------- |
> > | 0.08$\pm$ 0.01 | 0.44$\pm$ 0.02 | 0.11$\pm$ 0.02 | 0.48$\pm$ 0.03 |
> >
> > EGNN-H and SGNN-H are 4~5x lower than EGNN/SGNN.
> >
> >
> > > **Q4. Presentation in Related Work.**
> >
> > >**Authors refer to previous works such as Hamiltonian GNNs, and mention that they do not consider rotational equivariance. This is incorrect. In Hamiltonian GNNs, the edge embeddings can be modified to have the distance (L2 norm of the difference in positions), instead of giving simply the vectorial difference. Since the functional form to be learned in the case of a Hamiltonian is scalar, this approach also works very well and is both translationally and rotationally invariant.**
> >
> > Sorry for this mistake. We have revised the corresponding statements in the related work section. As mentioned in Q3, the experimental comparisons with Hamiltonian GNNs are also added.
> >
> >
> > > **Q5. Discussion of deformable systems.**
> >
> > >**However, in the present case although the particles are considered, there do not seem to be any deformation simulated and hence, these approaches referred should also stand equally valid as SGNN.**
> >
> > > **Although a particle-based approach is employed, unlike previous works such as GNS, there are no discussions on deformable systems**
> >
> > In fact, our model is generally capable of tackling those deformable objects. For the dataset Physion, the **Drape** scenario does require accurate simulation on deformable objects, which, in this case, refers to the cloth. The scene depicts the dynamics of cloth falling on some random objects. The experimental results clearly show that SGNN offers ~2% improvement in contact accuracy (c.f. Table 1) and ~10% lower MSE (c.f. Table 6 in Appendix) in this scenario. We have additionally contained a related video demonstration in the supplementary material.

---

> > > ### Author Response · Authors · 2022-08-02
> > > **Response to Reviewer cLqG (Page 3/3)**
> > >
> > > > **Q6. More baselines.**
> > >
> > > >**Also, the experiments chosen seem to be favorable for SGNN in comparison to the other baselines. For instance, implementing EGNN and GMN, exactly as they are, with gravity is expected to yield poor performance.**
> > >
> > > There is probably some misunderstanding here. We are **not** choosing the most favorable setting for SGNN but instead have tried our best to compare with the baselines fairly. **(1)** Our training and evaluation protocol strictly follow that of Physion [1] including their provided scripts of GNS and DPI; **(2)** For GNS and DPI, we also evaluate their data-augmented variants GNS-Rot and DPI-Rot by leveraging random rotations of the input data; **(3)** EGNN and GMN are equivariant models, and subequivariance is never investigated in these models. We involve them as baselines here to demonstrate the importance and necessity to relax equivariance to subequivariance for physical scene simulations, which is one of our core contributions.
> > >
> > > To further address the concern, we additionally design a $O_{\vec{\mathbf{g}}}(3)$-equivariant extension of EGNN and GMN. We achieve this by augmenting their update of the velocity as:
> > >
> > > $ \vec{\mathbf{v}}\_i^{l+1}=\phi\_v(\mathbf{h}^l\_i)\vec{\mathbf{v}}\_i^{l} + \underline{\phi\_g(\mathbf{h}\_i^l)\vec{\mathbf{g}}} + \sum\_{j\in\mathcal{N}(i)}(\vec{\mathbf{x}}\_i - \vec{\mathbf{x}}\_j)\phi\_x(\mathbf{m}\_{ij}), $
> > >
> > > where the underlined term is our modification and simulates the acceleration of gravity. We dub these two variants as EGNN-S ("S" for Subequivariance) and GMN-S, respectively. The results on Physion are depicted below:
> > >
> > > |        | Domino | Contain  | Link     | Drape    | Support  | Drop     | Collide  | Roll     |
> > > | ------ | -------- | -------- | -------- | -------- | -------- | -------- | -------- | -------- |
> > > | EGNN   | 61.3     | 66.0     | 52.7     | 54.7     | 60.0     | 63.3     | 76.7     | 79.8     |
> > > | EGNN-S | 72.0     | 64.6     | 55.3     | 55.3     | 60.5     | 69.3     | 79.3     | 81.6     |
> > > | GMN    | 54.7     | 57.6     | 54.5     | 57.6     | 55.1     | 54.2     | 79.5     | 81.3     |
> > > | GMN-S  | 55.6     | 65.3     | 55.3     | 57.0     | 59.3     | 57.3     | 81.2     | 79.3     |
> > > | SGNN   | **89.1** | **78.1** | **73.3** | **60.6** | **71.2** | **74.3** | **85.3** | **84.2** |
> > >
> > > We have the following observations:
> > > **(1)** Our designed $O_{\vec{\mathbf{g}}}(3)$-equivariant version of EGNN and GMN generally performs better than their $E(3)$-equivariant counterparts, which, again, indicates the necessary of leveraging proper symmetry constraint.
> > > **(2)** Even with these ad-hoc modifications, EGNN-S and GMN-S are still inferior to our SGNN by a large margin. This verifies the efficacy of our overall architecture.
> > > The above explanations and results have been added to Appendix.
> > >
> > > [1] Bear et al. Physion: Evaluating physical prediction from vision in humans and machines. NeurIPS 2021.
> > >
> > > > **Q7. The choice of the compared methods**
> > >
> > > >**More interesting examples where the situations where GNS and EGNN have been shown to yield SOTA performance could give a more realistic representation of how better SGNN is in comparison to these models.**
> > >
> > > We have already been comparing with the SOTAs on both datasets: GNS and DPI are indeed the SOTA simulators on Physion (c.f. Physion [1]), and DPI is the SOTA on RigidFall (c.f. [2]).
> > >
> > > [1] Bear et al. Physion: Evaluating physical prediction from vision in humans and machines. NeurIPS 2021.
> > > [2] Li et al. Visual grounding of learned physical models. ICML 2020.
> > >
> > >
> > > > **Q8. On self-contact.**
> > >
> > > >**Since the contact is of main concern in the present work, authors should evaluate how the model performs in situations where there is self contact.**
> > >
> > > > **Discussions on self-contact and how this is incorporated is missing.**
> > >
> > > As mentioned before in Q5, the `Drape` scenario in Physion is a deformable system and there are internal elastic forces and self-contact within the cloth themselves. Our model still performs promisingly in that scenario.
> > >
> > > > **Q9. In the case of a deformable system, consider a scenario where a particle from a given object breaks away, gets reflected by the wall, and then comes back to interact with the same object. In such case, if the particle comes with the $\varepsilon$ cutoff distance, does SGNN model this as a contact or as a particle of the object.**
> > >
> > > If the particle comes with the $\varepsilon$ cutoff distance, SGNN does **not** model this as a contact between different objects, but instead would model it as a particle of the object, since the particle's object label keeps unchanged. This interaction will be processed in the inner-object message passing stage in our hierarhical modeling.
> > >
> > > > **Q10. Performance of SGNN on systems with drag and other dissipative forces are lacking.**
> > >
> > > Actually, the systems in Physion are widely interfered by **Friction**, a typical dissipative force. Please refer to our demo videos in the supplementary material for better illustration.

---

> > > > ### Comment · Reviewer_cLqG · 2022-08-07
> > > > **Thank you for the response (part 3)**
> > > >
> > > > > If the particle comes with the $\varepsilon$ cutoff distance, SGNN does not model this as a contact between different objects, but instead would model it as a particle of the object, since the particle's object label keeps unchanged. This interaction will be processed in the inner-object message passing stage in our hierarchical modeling.
> > > >
> > > > I am not sure if the authors understood the question correctly. Consider a scenario where a bullet impacts a plate. In this case, part of the plate get broken and moves away from the original plate. My question was that if this plate comes back and hits the original plate, will this interaction be modeled as a contact, or will the plate automatically *heal*. As per authors' response it seems that, the broken part heals and gets attached to the original plate as this interaction will be processed in the inner-object message passing. **I believe this is a problem.** Because a broken object getting treated with the same interactions as the original object leads to unrealistic *healing*. I don't think the authors have addressed this point. However, this is a minor problem and can be addressed as part of a future study.

---

> > > > > ### Author Response · Authors · 2022-08-08
> > > > > **Further Responses (Part 3)**
> > > > >
> > > > > > **I believe this is a problem. Because a broken object getting treated with the same interactions as the original object leads to unrealistic healing.**
> > > > >
> > > > > Sorry for the misunderstanding, and thank you for the detailed explanations. We agree that this is a limitation, and have included this in the paper. We will keep on investigating potential solutions to this problem as future work.
> > > > >
> > > > > *Thanks again for your insightful suggestions. Please do not hesitate to contact us if there are other clarifications we can offer.  We would really appreciate it if you could **raise your rating**.*
> > > > >
> > > > > *Thank you for your time!*
> > > > >
> > > > > Best, \
> > > > > Authors

---

> > > > > > ### Comment · Reviewer_cLqG · 2022-08-08
> > > > > > **Thank you for the response**
> > > > > >
> > > > > > Thank you for the detailed response and for sharing additional results. I think these are interesting and worth discussing. However, I still share some of the original concerns and some concerns raised by other reviewers regarding limited applicability and incremental work from EGNN in terms of novelty. As of now, I shall keep the score as it is. However, I'll keep these additional points mentioned in the review response and consider it in the reviewer-meta-reviewer discussion period.

---

> > > ### Comment · Reviewer_cLqG · 2022-08-07
> > > **Thank you for the response (part 2)**
> > >
> > > > We employ a RK1 integrator to conduct Hamiltonian update.
> > >
> > > It is not clear to me why is RK1 used as the integrator? RK1 is not time reversible, non-symplectic and hence is not energy conserving. Isn't kind of contradicting when a non-energy conserving integrator is used along with the Hamiltonian equations? I noticed that several people have applied this in the literature as well. However, it is not clear to how this choice is justified. I am not expecting any additional experiments here. But I would appreciate if the authors can comment on this.

---

> > > > ### Author Response · Authors · 2022-08-08
> > > > **Further Responses (Part 2)**
> > > >
> > > > > **It is not clear to me why is RK1 used as the integrator?**
> > > >
> > > > We basically employ RK1 to control the computational cost. As depicted in the experiments in our previous response (please see Q3), even adding Hamiltonian update with RK1 integrator during training has already induced ~5x computational overhead. By further incorporating higher-order integrators like RK4 or those symplectic integrators, the cost will further increase by multiple times, making it hard to scale on large systems like those in Physion. As also mentioned by the reviewer, there are also works leveraging RK1 during training. Considering the trade-off between numerical precision and computational overhead, here we adopt RK1 during training in our previous experiment in the discussion.

---

> > ### Comment · Reviewer_cLqG · 2022-08-07
> > **Thank you for the response**
> >
> > > For example, the Hamiltonian-based NNs are generally designed to pursue energy conservation, which, however, is usually broken by forces like friction between objects, for the case like Physion. In this sense, the way we add inductive bias into SGNN is well appropriate.
> >
> > I understand that the authors do not aim to incorporate physical laws explicitly and aim to learn more from a data driven perspective. However, the above statement is only partially correct. While the general form of Hamiltonian equations are applicable for energy conserving systems, the equations can be easily modified to incorporate dissipation by adding an additional term in the second equation as $\dot{p}=-\partial H/\partial q + g(q,p)u$ (see: Gruver et. al., ICLR 2022), where $g(q,p)u$ is additional forcing term with $u$ being the control parameter. For instance, in the case of a linear drag, the forcing term will be $c\dot{q}$. Thus, frictional force can be made learnable in an HNN framework.
> >
> > >The video demonstrations were also provided in the supplementary material, where SGNN did produce visually realistic trajectories.
> >
> > While video demonstrations show good visualizations, they are only qualitative in nature. Rollout MSE is reasonable for non-chaotic systems. I would rely more on conserved quantities such as energy, force equilibrium, to show the trajectory is realistic.

---

> > > ### Author Response · Authors · 2022-08-08
> > > **Further Responses (Part 1)**
> > >
> > > We are truly thankful for your constructive comments. We really enjoy communicating with you and appreciate your efforts. We provide detailed explanations to your raised questions below.
> > >
> > >
> > > > **I understand that the authors do not aim to incorporate physical laws explicitly and aim to learn more from a data driven perspective. However, the above statement is only partially correct. While the general form of Hamiltonian equations are applicable for energy conserving systems, the equations can be easily modified to incorporate dissipation...**
> > >
> > > Sincerely thanks for the correction. We agree with the reviewer that Hamiltonian NNs can be modified to take into consideration dissipative systems and will revise the corresponding statement. We also thank the reviewer for pointing out this point, enlightening us to incorporate Hamiltonian into the SGNN framework for promising future work. In the current state, we identify several challenges to adapting Hamiltonian into physical scene simulation tasks like Physion:
> > > * The dissipative term $g(p, q)u$ [1] pointed out by the reviewer requires careful adaptation to this task. Particularly, other than being trivially implemented as MLP, there are several important inductive biases to be considered. For example, if $g(p, q)$ is for modeling forces exerted on the object level (like drag), it would be necessary for $g(p, q)$ to take into consideration the effect of multiple particles instead of a single one. In this case, $g(p, q)$ would probably be a GNN with multiple steps of message passing as well. Besides, if $g(p, q)$ is for modeling frictions, it might also be equivariant (or more precisely, subequivariant), since the generation of friction also follows physical symmetry.
> > > * Computational overhead. The Hamiltonian update requires taking the gradient of $H$ w.r.t. $p$ and $q$, as well as leveraging certain integrators to ensure desirable numerical precision. These operations significantly add up to the heavy computational cost for both training and inference, making it less scalable especially on large systems like Physion.
> > > * Different motivation. In many real-world scenarios, physical quantities like $p$ and $q$ may not be provided or even unmeasurable by visual perception (like cameras). Our goal here is to design a high-fidelity dynamics simulator with 3D information $\vec{\mathbf{x}}$ as input.
> > >
> > > To summarize, we agree with the reviewer that incorporating Hamiltonian is an interesting aspect to further investigate, and we also appreciate the reviewer for the fruitful suggestions on this topic. Nevertheless, injecting Hamiltonian would pose new challenges on this task, and our preliminary experiments (in Q3) in the previous discussion also verify that simply building up Hamiltonian into EGNN and SGNN does not lead to a desirable gain in performance. We leave this for future work and have added this point in Section 5 of the paper.
> > >
> > >
> > > [1] Gruver et al. Deconstructing the Inductive Biases of Hamiltonian Neural Networks. ICLR 2022.
> > >
> > > >**While video demonstrations show good visualizations, they are only qualitative in nature. Rollout MSE is reasonable for non-chaotic systems. I would rely more on conserved quantities such as energy, force equilibrium, to show the trajectory is realistic.**
> > >
> > > Thanks for the comment. Firstly, we would like to mention that the evaluation metrics we adopt in this paper are exactly those endorsed by the original benchmarks [2, 3].
> > >
> > > The reviewer raises a great suggestion for plotting the time evolution of certain quantities.
> > > For this purpose, we additionally compute the total energy (kinetic energy + gravitational potential energy) for those simulated Dominoes systems displayed in our demo video. To facilitate viewing, the resulting figures are provided in `energy.pdf` in this [anonymous link](https://drive.google.com/drive/folders/1NRBfwNk9yLMNii88ep0c0U8GquMKQeUb?usp=sharing).
> > >
> > > We interestingly find that SGNN closely tracks the value of the ground-truth energy, achieving the lowest error compared to all baselines.
> > > This verifies that the simulated trajectories are not only visually reasonable, but also physically valid. Particularly, SGNN simulates well even in those intervals when system energy changes drastically (e.g., due to the inelastic collision between Dominoes). It is also worth noticing from the figures that the systems in Physion are typically dissipative owing to friction or collision.
> > >
> > >
> > > [2] Bear et al. Physion: Evaluating physical prediction from vision in humans and machines. NeurIPS 2021.
> > >
> > > [3] Li et al. Visual grounding of learned physical models. ICML 2020.

---

### Official Review · Reviewer_tCa3 · 2022-07-19

**Rating:** 6
**Confidence:** 3
**Soundness:** 3 good
**Presentation:** 3 good
**Contribution:** 3 good

**Summary:**

This paper targets an interesting question of embedding equivariance into graph neural networks for physical dynamics recovery. The main focus is to 1) relax the strict constraint for cases with gravity and 2) consider the differences in self- and mutual interactions during learning.

**Questions:**

The review has some questions about the work.
1.	In Sec. 3.1. the authors define the problem target as to predict the position of the next time step (x^{t+1), where the position of the current time step is input (x^t). The reviewer would like to know if the proposed method is generalizable to a broader problem setup. There are many physical systems whose modeling can be seen as a mapping from one input quantity to one output quantity (xy). The graph structure is also applicable. How to generalize the proposed method then?
2.	The structure of the proposed Subequivariant Graph Neural Networks needs to be clarified. The reviewer suggests adding a flow chart around Sec. 3.2 -3.3.



**Limitations:**

Please see the questions above.

**Strengths And Weaknesses:**

This work targets the learning of physical system dynamics, which is an important topic and needs more attention in the community. The authors provide a clear illustration of the relevance to existing works on 1) graph neural networks to model interactions among particles/objects and 2) using physical constraints as an inductive bias to improve generalizability. The authors mainly propose subequivariance against the case with gravity and design multi-stage modeling to account for differences in object properties such as shape. To the best of the review’s knowledge, this work is new.

---

> ### Author Response · Authors · 2022-08-02
> **Response to Reviewer tCa3**
>
> We sincerely appreciate the reviewer's constructive comments and the recognition of the novelty of our paper!
>
> > **Q1. In Sec. 3.1. the authors define the problem target as to predict the position of the next time step ($x^{t+1}$), where the position of the current time step is input ($x^t$). The reviewer would like to know if the proposed method is generalizable to a broader problem setup. There are many physical systems whose modeling can be seen as a mapping from one input quantity to one output quantity ($x\rightarrow y$). The graph structure is also applicable. How to generalize the proposed method then?**
>
>
> We thank the reviewer for bringing up this interesting point. Our SGNN model is generalizable to a broader problem setup. To show this, let the input be a geometric graph of the system $\mathcal{G=(V(\vec{\mathbf{Z}}, \mathbf{h}), E, C)}$ with the node $i\in\mathcal{V}$ possessing some geometrical (vectorial) quantities $\vec{\mathbf{Z}}_i$ (e.g., position $\vec{\mathbf{x}}_i$, velocity $\vec{\mathbf{v}}_i$) and some scalar quantities $\mathbf{h}_i$,
>
> the edges $\mathcal{E}$ representing interaction or connectivity, and the object index function $\mathcal{C}$ as defined in the paper.
>
> Given the input, the model transforms as
> $$\vec{\mathbf{Z}}', \mathbf{h}' = f_{\text{SGNN}}\left( \vec{\mathbf{Z}}, \mathbf{h}, \mathcal{E}, \mathcal{C}\right).$$
> In this way, this formulation well covers what the reviewer has mentioned (i.e., "mapping from one input quantity to one output quantity"), and even generalizes it by allowing for both tensorial and scalar quantities. The dynamics prediction task in this paper is a specific example by setting $\vec{\mathbf{Z}}'=[\vec{\mathbf{x}}^{t+1}]$ with the input $\vec{\mathbf{Z}}=[\vec{\mathbf{x}}^{t}, \vec{\mathbf{v}}^{t}]$.
>
> > **Q2. The structure of the proposed Subequivariant Graph Neural Networks needs to be clarified. The reviewer suggests adding a flow chart around Sec. 3.2 -3.3.**
>
> We appreciate the reviewer's suggestion. Actually, we did include a detailed architectural flowchart of the message passing in Figure 8 of Appendix A.3. Combining the flowchart of message passing and that of the hierarchical design (Figure 2) would provide the necessary details of the entire SGNN model, which is additionaly depicted in Appendix in the revision.

---

> > ### Comment · Reviewer_tCa3 · 2022-08-08
> > **Thanks for the point-to-point response!**
> >
> > Thank you for answering the questions on generalizing modeling. Here is a follow-up comment.
> >
> > The reviewer suggests 1) highlighting the key differences and key design of SGNN in Fig. 8, and 2) referring to Fig. 8 in the appendix when illustrating the proposed GNN structure.

---

> > > ### Author Response · Authors · 2022-08-09
> > > **Thanks for your helpful suggestion!**
> > >
> > > Dear Reviewer tCa3,
> > >
> > > Thank you very much for the detailed suggestion! We have modified Fig. 8 to highlight the key components by red circles. We have also revised the manuscript in Sec. 3.2 to refer the readers to Fig. 8 for the detailed illustration of the proposed structure.
> > >
> > > We enjoy the discussions and thank you again for these helpful comments that have greatly improved the paper!
> > >
> > > Best, \
> > > Authors

---

### Official Review · Reviewer_Y6Jq · 2022-07-26

**Rating:** 7
**Confidence:** 4
**Soundness:** 3 good
**Presentation:** 2 fair
**Contribution:** 3 good

**Summary:**

The manuscript improves the equivalent graph neural network (GNN) to tackle the possible inefficiency when the model predicts the dynamics of a physical system when the system symmetry is partially broken by an external force such as gravity.  The subequivariant GNN (SGNN) proposed in this work introduces the hierarchical architecture in terms of the particles and objects where the subequivariant message passing is incorporated into the model to deal with complicated object interactions. The approach with the additional freedom can be more accurate in the task to evaluate the physical dynamic of objects from vision and achieve an impressive generalization compared with GNS model etc.

**Questions:**

Q1: As mentioned previously, the author should make a better introduction to make people easy to understand the background and motivation as well as the challenge of the task.

Q2: It seems the work is similar to the development of the machine learning force field (MLFF) in molecular dynamic simulation. Probably, the task predicting the physical dynamic of the objects has less information from the vision system than that can be obtained in an atomic system. It would be better if the author could clarify the difference between the two tasks.

Q3: If the SGNN mode is capable of predicting the force field of the atomic system, it would be interesting to compare with the MLFF models. There are many recently developed models such as Nequip and TorchMD incorporating the rotation equivariance and achieving impressive high accuracy while predicting the atomic force field.

Q4:  Besides Physion, is there any other dataset for further verifying the performance of the model? Unlike SGNN, it seems the compared models, such as GMN and EGNN are general models and not specifically developed for this task. The comparison may not be fair. It would be better to compare with the model which is dedicated to this task. Otherwise, the SGNN should be compared with other models in different tasks.


**Limitations:**

Yes. This is a work dedicated to developing the machine learning model for science. There would be no potential negative societal impact.

**Strengths And Weaknesses:**

Originality:  The work introduces the concept of subequivariance based on the hierarchical network to extend the capability of GNN in the task of learning the physical dynamics from vision information. The work has very good originality.

Quality: The model is developed rigorously based on the understanding of the rotation equivariant group. The design and test of SGNN with Physion dataset are thoughtful and convincing.

Clarity: The presentation of the manuscript is clear. However, the background and the motivation are confusing. It is a bit hard to understand the challenge of the problem until I read the online introduction of Physion dataset.

Significance: Though I may not consider this work as a breakthrough of GNN model development in the focused task, the fresh idea, such as the subequivariance described in the manuscript, indeed helps the neural network achieve better accuracy and generalization. The work can bring people to pay attention to the importance of developing a model with appropriate physical constrain for a similar task.

---

> ### Author Response · Authors · 2022-08-02
> **Response to Reviewer Y6Jq (Page 1/2)**
>
> We thank the reviewer for the detailed comments and the recognition of our paper!
>
> > **Q1: As mentioned previously, the author should make a better introduction to make people easy to understand the background and motivation as well as the challenge of the task.**
>
> Thank you for the advice. We agree with the reviewer and apologize for not elaborating and emphasizing the challenges of the task sufficiently. We have carefully made revisions to the manuscript to properly reflect the challenges, including the motivation and background. Details are provided in the Introduction.
>
> > **Q2: Probably, the task predicting the physical dynamic of the objects has less information from the vision system than that can be obtained in an atomic system. It would be better if the author could clarify the difference between the two tasks.**
>
> Thanks. We recognize that the task of modeling the physical dynamics of objects in vision systems (e.g., Physion) poses **unique challenges** over the atomic systems in various aspects.
>
> **1.** The objects in these systems are diverse in material (e.g., bowls, bricks, dominoes, cloth) and hardness (rigid or deformable). These physical properties significantly influence the way the objects interact, posing great challenges to the generalization of the simulator. We approach this challenge by taking into consideration the object's geometrical information. However, in atomic systems, these concepts become minor where the focus mainly transits to atoms themselves and the force field between atoms.
>
> **2.** There are multiple types of interaction forces in these systems, and these forces usually have substantial differences from each other. Typical interactions include collision (where objects fiercely exchange momentums), friction on contact surfaces, and even internal forces that help maintain the shape of objects. As supported by the experiments on all 8 scenarios in Physion, our model is able to capture a broad range of interactions.
>
> **3.** Physical scenes also usually involve external forces, which, in Physion dataset, corresponds to gravity. This factor motivates us to design subequivariant models by incorporating the external force into message passing, ensuring desirable symmetry. As for the atomic systems, molecules are mostly simulated in vacuum or implicit solvent, where such consideration of external force has not been elaborated.
>
> **4.** The systems considered in this paper are of much larger size than previously explored atomic systems such as QM9 [1] and MD17 [2]. Specifically, systems in Physion contain **thousands** of particles, while one molecule in QM9 and MD17 consists of no more than 100 atoms. When modeling large systems like Physion, it is required to take care of the scalability of the proposed method.
>
> [1] Ramakrishna et al. Quantum chemistry structures and properties of 134 kilo molecules. Scientific data. 2014.
> [2] Chmiela et al. Machine learning of accurate energy-conserving molecular force fields. Science advances. 2017.

---

> > ### Author Response · Authors · 2022-08-02
> > **Response to Reviewer Y6Jq (Page 2/2)**
> >
> > > **Q3: If the SGNN model is capable of predicting the force field of the atomic system, it would be interesting to compare with the MLFF models.**
> >
> > We are happy that the reviewer thinks our model is general and has the potential to be applied to atomic systems.
> >
> > However, the core interest of our paper here focuses on learning physical dynamics. As already explained in Q2 above, the diversity of the objects and the existence of gravity motivate the specific design of SGNN (involving object-level message, considering subequivariance, and using the hierarchical framework). In the current version, our SGNN model is particularly designed for the macro physical systems in our daily life, instead of the atomic systems, where special featurizers (e.g., radial basis functions [3,4]) are commonly applied. We will also release our code for the community to develop upon SGNN on other interesting tasks.
> >
> > [3] Schütt et al. SchNet: A continuous-filter convolutional neural network for modeling quantum interactions. 2017.
> >
> > [4] Klicpera, Johannes et al. Directional message passing for molecular graphs. ICLR. 2020.
> >
> > > **Q4: Besides Physion, is there any other dataset for further verifying the performance of the model? Unlike SGNN, it seems the compared models, such as GMN and EGNN are general models and not specifically developed for this task. The comparison may not be fair. It would be better to compare with the model which is dedicated to this task. Otherwise, the SGNN should be compared with other models in different tasks.**
> >
> > Indeed, besides Physion, we have conducted evaluations on RigidFall (please see Table 3 and Figure 6). GNS and DPI have been reported to be State Of The Art (SOTA) on Physion (see [5]), and DPI is SOTA on RigidFall as well (see [6]). Therefore, we have indeed compared the most competitive baselines in the literature, in addition to EGNN and GMN. Moreover, we also endeavor multiple ways to further boost the performance of these baselines. Even so, SGNN still yields significant improvement over SOTAs. It is true that EGNN and GMN are not particularly designed for large-scale physical scene simulations, but we still include these equivariant models as baselines in order to verify the validity and effectiveness of our proposed components. As also recognized by the reviewer, one purpose here is to emphasize that proper physical inductive biases should be carefully considered when designing GNN simulators on scene tasks.
> >
> > We are happy to provide additional results if the reviewer thinks there is still any experiment missing on learning physical simulation.
> >
> > [5] Bear et al. Physion: Evaluating physical prediction from vision in humans and machines. NeurIPS 2021.
> >
> > [6] Li et al. Visual grounding of learned physical models. ICML 2020.

---

### Author Response · Authors · 2022-08-02
**General Response**

We sincerely thank all reviewers and ACs for their time and efforts on reviewing the paper. We are glad that the reviewers recognized the contributions of our paper, which we briefly summarize as follows.

* **Novelty.** "The work has very good originality." "The work can bring people to pay attention to the importance of developing a model with appropriate physical constrain for a similar task."(Y6Jq) "To the best of the reviewer’s knowledge, this work is new."(tCa3)
* **Experiment.** "The design and test of SGNN with Physion dataset are thoughtful and convincing." (Y6Jq) "The experiments seem thorough and multiple baselines are used." (bh6f)
* **Presentation.** "The presentation of the manuscript is clear." (Y6Jq) "The authors provide a clear illustration of the relevance to existing works." (tCa3) "Overall, the work is well-written and clearly presented." (cLqG)

We also appreciate the reviewers for their thoughtful comments and concerns. Below we summarize two core aspects: **(1)** the main focus and contributions of the paper; **(2)** the revisions made to further improve the paper, including the extra experiments we have added.

**1. Main focus and contributions.**

In this work, we aim to tackle the physical simulation task on challenging physical scenes including 8 scenarios on Physion and RigidFall. Such tasks are **highly challenging**, due to **the scale of the system** (on average thousands of particles per system), **the diversity of the interactions** (e.g., collision, friction, gravity), as well as multiple **shapes, materials, or even rigidness** of the objects. Particularly, **both rigid and deformable objects** exist in Physion. We kindly recommend the reviewers watch our demo video in the supplementary for a better understanding of these summarized challenges of the task.


To this end, we propose a particle-based GNN simulator, SGNN, that injects mild inductive biases including physical symmetry, particularly when external force like gravity exists, and object geometric information into the hierarchical model. The model can well simulate both rigid and deformable objects, is capable of modeling diverse interactions, and is promised to meet the proper symmetry, without relying on specific physical PDEs like the Hamiltonian. We also theoretically reveal that our subequivariant message passing has universality guarantee.


**2. Summary of the revisions.**
We add the following experiments per the reviewers' suggestions.
* Reviewer cLqG suggests adding strong physical priors like Hamiltonian into the network. We augment EGNN and SGNN with Hamiltonian updates and add experimental comparisons with these models.
* Reviewer cLqG suggests stronger baselines designed on top of EGNN and GMN. We design the subequivariant version of EGNN and GMN, and add corresponding experiments.
* Reviewer bh6f asks about the difference between our subequivariant message passing and group representation-based steerable convolutions. We make non-trivial efforts to adapt the mentioned approach, which is implemented only on CNNs, to GNNs and compare this baseline with our model.

The results of all these experiments still demonstrate that SGNN offers significant improvements in all scenarios compared with these baselines.

We have also added more discussions about physics-informed physical models and the steerable E(2) CNNs, along with adding citations to the relevant works mentioned by the reviewers.

---

### Author Response · Authors · 2022-08-06
**Thanks for all your comments and look forward to post-rebuttal feedbacks!**

Dear AC and all reviewers:

Thanks again for all of your constructive comments, which have helped us improve the paper!

Since the discussion phase started over three days, we have not heard any post-rebuttal response yet.

Please don’t hesitate to let us know if there are any additional clarifications or experiments that we can offer. We would love to discuss more if any concern still remains. We appreciate your suggestions.

Thanks!

---

### Meta-Review · Area_Chair_eqHZ · 2022-08-26

**Recommendation:** Accept
**Confidence:** Less certain

**Metareview:**

Overall this is an interesting paper. It proposed a new formulation of the equivariant graph neural network, subequivariant GNN. Reviewers agree that the proposed idea could be useful to the community, albeit with perhaps small application scope. So on the novelty side, this paper is okay. The biggest concern among the reviewers is about the experiments, i.e., mostly the fair comparision. I feel the authors did a reasonable job to explaining why the current baselines were chosen and provided additional experimental evidence. Authors could take in account the comments from the reviewers to improve the overall presentation of the paper.

**Award:**

No

---

### Decision · Program_Chairs · 2022-09-14

Accept